# Financial Crises and Business Cycle Implications for Islamic and Non-Islamic Bank Lending in Indonesia

**Samar Issa**

Department of Business Administration, Saint Peter's University, Jersey City, NJ 07306, USA;
sissa@saintpeters.edu

**Abstract:** This paper addresses if and how excess debt can be considered as an early warning signal for banks and takes an additional dimension by comparing the excess leverage between Islamic and conventional banks in Indonesia before, during, and after the Global Financial Crisis (GFC). To do so, this research develops an empirical model of banks' capital structure and optimal and excess debt, and it conducts an empirical analysis on the overleveraging of eleven conventional and Islamic banks in Indonesia. Results show that all banks became vulnerable to the GFC in 2007–2009 while credit build-up, i.e., overleveraging, started in 2005. However, for most of the banks in the sample, Islamic banks performed better and leveraged less prior to the GFC, which made them more resilient to the crisis.

**Keywords:** banking instability; banking sector; Islamic banking; credit flows; financial crisis; excess debt; early warning signals

## 1. Introduction

Over the past two decades, excessive capital borrowing (leveraging) has risen significantly for the financial sector globally. Following the 2008 Global Financial Crisis (GFC), there has been a rise in academic and policy research on the problems of debt sustainability. The COVID-19 pandemic has reignited the urgency of the topic. While more solid empirical assessments in relation to the pandemic are yet to come, this paper asks the question whether Islamic banks, which are argued to be safer than conventional banks for having their product structured as asset-backed financing, have maintained a more sustainable debt level pre- and post-GFC. Many studies investigated issues related to asset–price channels through which the banking system's instability is triggered. Some of these important academic contributions include Brunnermeier and Sannikov (2014), Mittnik and Semmler (2012, 2013), Stein (2012a, 2012b), and Gross et al. (2017).

To undertake the empirical study on those issues in Indonesia, this paper uses a measure of overleveraging as defined by Stein (2012a). Essentially, if borrowing exceeds debt capacity, then this can be called "excess leveraging". Debt capacity is measured as sustainable or optimal debt. In other words, the measure of overleveraging is defined as the difference between actual and optimal debt. The model presented in this paper follows Stein (2012b) and Issa (2020), and it focuses on the solution of the dynamic version of the Stein (2012b) model that allows us to use time-series data on banks in order to calculate the excess debt of eleven banks in Indonesia.

In this paper, overleveraging refers to lending booms. It is well known that lending booms may precede banking system instability, as they imply increased risk taking in the financial system. This has the potential to result in financial turmoil if the economy is hit by a negative adverse shock in asset prices as occurred during the GFC of 2008.

Banks have continued to leverage after 2012, and there are expectations of another financial crisis affecting the banking sector following the COVID-19 crisis; we have already seen the effect on the financial market. However, it is important to differentiate between

the nature of the external COVID-19 shock and the 2008 crisis. The former is a purely exogenous shock, while the latter is a result of a vulnerable state of the financial system. COVID-19 provides a shock that is not connected to business or financial cycles and, as such, behavior can be analyzed empirically with less concern for the usual endogeneity issues. It must be noted that the government should control leverage in order to prevent a future crisis of any type (see what happened in COVID-19). The COVID-19 pandemic delivered a structurally different impact but still layering upon subdued endogenous fragilities. For the main endeavor of this paper, this paper proceeds with the core Stein model as laid out in Equations (1) and (2), leaving the COVID-19 crisis and the extension opportunities for future projects.

How did the overleveraging of financial intermediaries occur in Indonesia? After the bust of the technology boom in 2000, not only asset prices rose globally but financial intermediaries increased their loan supply to the real estate sector. Since the 2008 GFC and subsequent quantitative easing by developed countries, large capital inflows into emerging market economies (EMEs) were induced. Many of the financial and real trends that occurred after 2001 are documented in Semmler and Bernard (2012) and Stein (2011). The extent to which these countries are affected depends on each country's financial system and macroeconomic conditions. For instance, Indonesia is a bank-based economy characterized by shallow financial markets, which posits the economy at a higher volatility and instability. Moreover, global factors have played a role in adjusting the country's monetary policy through a number of channels such as the interest rate, exchange rate, asset prices, bank deposit and lending, and debt issuance as well as risk taking and excessive leverage.

This paper addresses if and how excess debt as defined by Stein (2012a) can be considered as an early warning signal for banks and takes an additional dimension by comparing the excess leverage between Islamic and conventional banks in Indonesia before, during, and after the GFC. Results showed that all banks became vulnerable to the GFC in 2007–2009, whereas credit build-up, i.e., overleveraging, started in 2005. However, for most of the banks in the sample, Islamic banks performed better and leveraged less during the crisis, which meant these banks were affected less by the crisis and also made them more resilient to a financial crisis. Nevertheless, Islamic banks still in some cases relied on leveraging and were subject to the second-round effect of the global crisis around the years of 2011–2013, when higher excess debt values were found for Islamic banks. That could also be due to the fact the financial crisis hit the real economy after 2009 because most Islamic bank investments and contracts were backed by real estate and property as collateral. The results show that the challenges in Indonesia became even more complex in 2013 as they coincided with the domestic problem of inflation pressure from an increase in domestic fuel prices and a widening current account deficit due to the decline in commodity prices together with booms in lending and house prices.

This paper is organized as follows. Following the introduction and rationale of this study in Section 1, Section 2 includes the literature review of Islamic banking in Indonesia. A theoretical model of optimal leveraging is developed in Section 3. Section 4 presents empirical estimation results with some interpretations, and Section 5 concludes the paper. The appendices provide the technical background and calculations of excess debt.

## 2. Literature Review

The expansion of Islamic banking has motivated several researchers to assess Islamic banks' performance. A recent study by Marlina et al. (2021) aimed to analyze the growing trend of available financial research regarding Islamic banking. The authors used bibliometric mapping and found that as of January 2021, there were 500 published scientific research papers, with journal articles being the great majority. They found that the most common authors were Sukmana, Ascarya, and Ismal. The keywords most often used were "Indonesian" and "Islamic". The newest trend topics circle around governance, sharia issue, and the social role of Islamic banking in Indonesia. As Islamic banking is growing at a fast

rate, the new research contains many beneficial findings that can be implemented to fix and improve issues within the financial industry and, in return, improve people's lives.

An International Monetary Fund (IMF) paper by Bitar et al. (2017) studied the effect of Basel Core Principle (BCP) compliance on the stability of conventional and Islamic banks. The authors used a sample of 761 banks in 19 countries for the years 1999–2013. Their findings suggest that the BCP index is positively associated with the stability of conventional banks, whereas the effect is less noticeable for Islamic banks.

Another IMF paper by Shabsigh et al. (2017) showed that the complexity of Islamic banks exposed them to additional risks that should be addressed by supervisory authorities and central banks. Islamic banks' hybrid products exposed them not only to additional returns but also to added risks that made them volatile. The authors argued that there was a need to establish a clear framework for troubled Islamic banks to ensure the stability of the financial system.

A number of recent theoretical studies have explored the Islamic banks in Indonesia specifically. For instance, Kusuma and Ayumardani (2016) investigated the relationship of bank performance with its corporate governance. The authors selected a sample of eleven Islamic banks for the years 2010–2014 and performed a regression analysis on the data extracted from the balance sheets of banks. The findings showed that the efficiency level of corporate governance of Indonesian Islamic banks improved during the period in context. In addition, corporate governance efficiency was significantly correlated to Islamic banks' performance.

A recent paper by Abduh and Omar (2012) took a macroeconomic approach and tried to find the correlation between Islamic banks' short-term and long-term performance and Indonesian economic growth. The methodology in this research used quarterly data from 2003 to 2010, extracted from the International Financial Statistics of IMF and from the monthly Islamic Banking Statistics Report of the Bank of Indonesia. The variables representing the Islamic banks were total financing, and the indicators representing the Indonesian economy were the Gross Domestic Product and Gross Fixed Capital Formation representing investments. The authors found a relationship between Islamic banking and Indonesian economic growth. However, the surprising aspect was that the relationship was bi-directional, meaning that Islamic banks promoted economic growth while economic growth itself also stimulated Islamic banking. Thus, both supply-leading and demand-following relationships were taking place at the same time. In order to use this information, the author suggested that the Indonesian government should incorporate Islamic banking more in Indonesia as well as review the current regulations because they have been shown to have no benefit with the corresponding growth.

Furthermore, Cakranegara (2020) attempted to analyze the severity of the damage caused by COVID-19 in the financial sector by comparing the COVID-19 crisis with the 1998 crisis. The purpose of the study was to identify similarities between the two crises that would allow regulators and managers to draw lessons from the past and apply it to the current crisis. The author claimed that Indonesia maintained stability through its system, consisting of "several layers" such as the debt selection system, the loan distribution system, the minimum capital owned by a bank, the rule to use asset insurance, the banks' health monitoring system, and government intervention when everything else failed failed. Therefore, Indonesian banks diversified risk through different industries, periods, and products in order to lower their systemic risk. Nevertheless, when the COVID-19 pandemic hit, every industry was hit and, thus, all banks were affected. The author stated that another approach is needed in order to prevent the same fall from happening in a future crisis. This paper found that Indonesian banks nowadays are far more resilient and that people also find them more reliable than they were in 1998. The lower effect of the crisis was due to the improved capital structure of the banks as well as the more lenient actions taken by the government. The author believed that strict government measures in 1998 contributed to the banking recession and stated that public trust and credit are key contributors to economic growth.

Maghfuriyah et al. (2019) similarly pointed out that short-term and long-term market structures affected banking performance in Islamic banking. To achieve that, the authors used the Structure Conduct Performance analysis by implementing an error correction model. They utilized monthly time-series data from 2015 to 2018. The data were acquired from the Islamic banking monthly report, the Islamic banking statistics published by the Financial Service Authority, and the financial statement of each Islamic bank in Indonesia. The market structure variables used in the study were the concentration ratio, the Herfindahl–Hirschman Index, and the market share. Concentration ratio was defined as "the percentage of the overall industrial output generated by the largest companies", and the Hirschman Index was used to obtain an accurate description of the analysis concentration ratio. The error correction model showed that in the short term, market structure had an insignificant effect on banking performance growth. However, in the long term, the opposite occurred; market structure had a great effect on banking performance due to certain determinants. For example, if the market share of deposits increased, then the Islamic banking performance would decrease in the long term but increase in the short term. Moreover, if the concentration ratio increased, then the Islamic banks performance would decrease both in the short and long term. On the other hand, if the market concentration ratio decreased, then the banking performance would increase. With such findings, the authors hoped that regulators and firms could make better informed decisions to allow the Islamic banks to flourish both in the short and long term.

The paper by Nugroho et al. (2018) showed how segmentation affected the quality of Islamic banks based on non-performing financing ratio/non-performing loan (NPF/NPL). The research used a quantitative method with multiple regression tests and demonstrated that retail segmentation held more weight than whole segmentation did in influencing the quality of the banks (i.e., 92.61% and 56.05%, correspondingly). Moreover, the study stated that Islamic banks faced greater challenges than conventional ones. In 2017, starters only held 10% of the market share, while conventional banks held 90%. Not to mention that some key financial indicators, such as low-quality assets, showcased the financing channeled by Islamic banks. Based on NPF Gross, roblematic financing in (the Islamic) banking industry grew by 341.85% over eight years from 2008 to 2015. The authors concluded that given their findings, it was critical that for the retail segment to find a "special treatment" to maintain the high quality of assets and Islamic bank performance. In other words, banks should reinforce and follow up with their sharia principles in contributing to social and environmental aspects to engage more customers with their services.

Another paper by Rodoni et al. (2017) focused on finding the efficiency rate and productivity of Islamic banks given their reliance on that to compete against conventional banks. The model used time-series secondary data extracted from sources such as bank reports, statements publications, and other relevant sources. The technique used by the authors to obtain the efficiency rates was the Data Envelopment Analysis and for productivity was the Malmquist Index (MI). The study refers to data from 2009 to 2013. The findings showed that most of the Islamic banks had inefficient rates. The Indonesian Islamic banks ranged between 87% and 97% in efficiency rates and never reached 100%, which the authors attributed to external rather than internal factors. Moreover, the authors stated that during the last four years, there has been a trend of increasing productivity but that such was caused by managerial factors rather than technological ones. As for Malaysian Islamic banks, the situation was somewhat better. Their efficiency rates ranged from 92% to 95%, even though in the last 5 years, they have not reached a 100% efficiency rate, either. The percentage of inefficiency was attributed to external factors as well. Lastly, there has been a fluctuating and a likely negative growth in productivity that is mostly due to technological issues.

Third, Pakistani Islamic banks have performed better compared to the Indonesian and the Malaysian banks. The efficiency rate varied between 99.3% and 100% during the last 5 years. As for productivity, their trend has been positive and increasing due to technological advancements and good management.

Mulyaningsih and Daly (2011) analyzed the competitiveness of the Indonesian banking industry based on consolidation and concentration. The authors claimed that in 1990, banks experienced a structural change due to the deregulation policy in 1988, making it easier to enter the banking market. However, in 1995, the government started to tighten the regulations once again. In 1997 (post-1995 crisis), the Central Bank merged four state-owned banks and closed the operation of twenty-three others in order to achieve economies of scale and healthier systems. To increase stability in the banking industry, the Central Bank focused on increasing monitoring based on international standards and fostering good corporate governance. To assess the competitiveness of the banks, the authors used a regression model. The authors found that the structure of the Indonesian market was vulnerable, given that the market was concentrated in a few banks because large banks control a substantial share. Second, the consolidation actually led to a less concentrated market and better market share. Third, during the consolidation policy, banks were running under "monopolistic competition" for large banks, but after some time, competition started growing for all. The authors showed that a concentrated market led to less competition, which explained why larger banks in Indonesia who had higher concentration also had less competition.

Empirically, a paper by Asutay and Izhar (2007) analyzed the performance (profitability) of Bank Muamalat in Indonesia. The authors' methodology relied on a regression analysis to estimate the internal and external determinants in reference to its return on assets. The research used monthly bank-specific data extracted from balance sheets and income statements from 1996 to 2001, while macroeconomic indicators were derived from the Bank of Indonesia. The authors found that the bank's profit largely came from financial activities and not from service activities. Moreover, from 1996 to 2001, the bank's portfolio relied on too much short-term-based financing. A positive correlation between the ratio of total liabilities to total assets showed that Islamic banks have incentives to take on more risks. Nevertheless, a negative coefficient of this same ratio showed that Islamic banks tended to lose money when they offered more of the Mudarabah schemes. The authors' findings also matched with the hypothesis that there is a direct relationship between inflation and profitability measure.

### 3. Empirical Approach and Definition of Variables

*3.1. Background of Financial Crisis and Islamic Banking*

According to many researchers, the GFC crisis was caused by excessive financial obligations/mortgages of private households (i.e., bubbles in the mortgage market defined as unsustainable debt/income ratios). Stein (2012b) argued that although debt problems may have originated in either the public or private sectors in different nations, the result was still declining asset values, and the mechanisms at work resulted in a contagion effect either from the United States to Europe and/or from one European nation to another, depending on the debtor-to-debtee relationship under examination. Of course, in each scenario, Stein made it clear that the primary source of the problem was not the presence of debt but excess debt within the country/countries under analysis.

Stein derived an optimal debt ratio and built on it to identify an early warning signal (EWS) of a debt crisis, which is defined as the "excess debt of households" (actual debt ratio less than the optimal debt ratio). As the excess debt level rises, the probability of a debt crisis increases. It has been shown that rising house prices since the late 1990s led to above-average capital gains for households, thereby increasing owner equity. The supply of mortgages increased and, consequently, financial obligations as a percentage of disposable income increased for private households. At the same time, the quality of loans declined (subprime mortgages). Of course, this process was not sustainable. As capital gains[1] dropped below the interest rate, debtors could no longer service their debts and foreclosures led to a collapse in the value of financial derivatives.

Before delving into the theoretical model and empirical analysis presented in this paper, it is important to briefly provide background information regarding Islamic banks'

mortgage-loan operations. As mentioned, their operations do not involve interest rates; rather, they have a proprietary program called the LARIBA (interest-free) model, which uses equity-participation systems or profit/loss sharing. Islamic banks offer two types of mortgage loans. First, profit sharing (murabaha) is offered whereby the bank does not loan money to the buyer to purchase the home or other property; rather, the bank buys the home itself and then resells it to the buyer at a profit. The buyer typically pays a fairly large down payment, and the price at which the bank buys the property is disclosed to the end buyer. The second method is decreasing rent (ijara). The bank purchases the home and resells it to the buyer; however, unlike the first method, in this case, the home remains in the bank's name until the total price is paid. The buyer takes up residence immediately and makes payments to the bank on the purchase price, but in addition to the payments, the buyer also pays a fair market rent. This method is preferred in countries such as the United Kingdom to avoid the double payment of taxes (Khan 2010).

Dr. Abdul-Rahman, Chairman and CEO of the Bank of Whittier, said during an interview that they had few nonperforming mortgages in 2008 and that this was due to their riba-free (RF) discipline (Abdul-Rahman 2010). He explained that the bank tracks home prices in US dollars relative to more stable commodities, such as gold, silver, wheat, or rice, thereby allowing the discovery of real estate bubbles. He added that on the basis of the used strategy, the bank had an early warning signal of a macro-bubble forming in the real estate sector. In addition, they matched the property to the market, meaning that they researched the rental value of a similar property in the same neighborhood. He stated that the inputs of their model are the amount to be invested (financed), the number of years of financing, and the rent. The unknown is the rate of return on investment[2]. Their internal business model allowed them to survive the recent financial crisis.

### 3.2. Theoretical Model

This paper introduces a model of optimal leverage that helps us define overleveraging. The model sketched here is a low-dimensional stochastic variant of a model of banking leveraging, and it follows Stein (2008, 2012a) and Brunnermeier and Sannikov (2014).

This paper focuses on the Stein (2008, 2012a) model. However, several recent studies in the literature develop models to estimate excess debt, for both private and public debt, and many of them stem from the same (Stein 2008, 2012a) Stein model.

For instance, Issa and Gevorkyan (2022) develop an empirical model of corporate capital structure, optimal debt, and overleveraging, covering approximately the period from 2000 to 2018, across the following six industries: technology, financial, pharmaceutical, auto, airline, and energy. The authors estimate optimal and excess debt for each of the 89 firms and find that corporate excess debt has largely been moving up, spiking around the GFC and continuing into recovery more recently. The trend is consistent with an increase in the actual debt, with varying average excess debt ratios by sector.

An IMF paper by Eberhardt and Presbitero (2018) develops an empirical model using regression analysis to predict banking crises. The authors employ a sample of 60 low-income countries for the time span 1981–2015 and show that credit growth cannot be considered a main driving force of economic distress or financial crisis, as most of the literature states. The reason is that credit growth is mediated through capital inflow, which is also fueled by a booming financial market. Therefore, banking crises are potentially driven by commodity price changes according to the authors.

Another paper by Nyambuu and Bernard (2015) uses the Stein (2008) model to calculate optimal debt for developing countries. In contrast to this paper, which calculates the optimal debt at the micro-level for the baking sector, the authors apply the model at a macro-level. In both cases, the optimal debt ratios are defined as the threshold beyond which it becomes risky to borrow, or in other words, the distance-from-default indicator variable. While both papers calculate optimal debt, each methodology applies different metrics, since this paper addresses banking default while Nyambuu and Bernard (2015) report sovereign default. Nyambuu and Bernard (2015) show that rising ratios of external

debt can make a country vulnerable to shocks and increases the risk of default. Similarly, this paper shows that an increase in excess debt of a certain bank will increase its risk to default on its debt.

Finally, Gevorkyan and Semmler (2016) examine the U.S. shale energy sector, and their model is consistent with Stein's optimal debt argument as well. The framework is rooted in the theory that the boom trend in borrowing leads to overleveraging and risks of insolvency. Employing a nonlinear model predictive control (NMPC) analysis, the authors present two mutually exclusive scenarios: oligopoly or extensive competition. The empirical analysis of actual and optimal debt based on a vector error correction model (VECM) suggests that firms with larger market capitalization in the industry are more resilient to oil price volatility than medium and small-sized firms.

Overall, this model is very similar to those of Brunnermeier and Sannikov (2014) and Stein (2012a). Both models have leveraging and payouts as choice variables and net worth as a state variable. Moreover, both models are stochastic. Similar to this study, Brunnermeier and Sannikov (2014) specifically focused on the banking sector; however, the setting is more general compared to the one used in this paper. There are households that save and financial experts representing financial intermediaries that invest in capital assets owned by households and financial intermediaries. Both have different discount rates. I focus solely on the behavior of financial intermediaries.

In this model, the preferences used in the objective function and Brownian motions as state variables are similar to those in both studies. The Stein (2012a) model, assuming certain restrictions, uses log utility and allows exactly computing excess leveraging. Capital return is also stochastic due to capital gains, and the interest rate is stochastic as well, similar to my model and in contrast with that of Brunnermeier and Sannikov (2014) where only the capital return is stochastic, and the interest rate is taken as constant. Both Brunnermeier and Sannikov (2014) and Stein (2012a) employed a continuous time version, but the problem in this paper is formulated as a discrete time variant with a discounted instantaneous payout and an optimal leveraging function.

Moreover, a shock to asset prices creates a vicious cycle through the balance sheets of banks. In other words, risk taking and excessive borrowing occur when asset prices are volatile. In this model, the unconstrained growth of capital assets through excessive borrowing, facilitated by the lack of regulations imposed on financial intermediaries, is considered the main cause for banking sector's instability[3]. On the other hand, large payouts with no "skin in the game" affect banks' risk-taking behaviors, equity development, and leveraging. The higher the payout is, the more leveraged the bank becomes, which increases the aggregate risk and risk premia for all. In summary, the increased risk spreads, and risk premia, especially at a time when defaults begin, expose banks to vulnerabilities and financial stress triggered by security-price movements.

To derive an optimal debt ratio, Stein used stochastic optimal control (SOC). A hypothetical investor selects an optimal debt ratio, $f(t)$, to maximize the expectation of a concave function of net worth, $X(t)$, where $t$ is the terminal date. The model assumes that the optimal debt/net worth ratio significantly depends on the stochastic process concerning the capital gain variable. The expected growth of net worth is also maximal when the debt ratio is at the optimal level.

Optimal leverage is given by:

$$f^*(t) = \left[(r - i) + \beta - \alpha y(t) - \frac{\left(\frac{1}{2}\right)\left(\sigma_p^2 - \sigma_i \sigma_p \rho\right)}{\sigma^2}\right] \tag{1}$$

such that

$$Risk = \sigma^2 = \sigma_i + \sigma_p - \left(2\rho_{ip}\sigma_i\sigma_p\right), \tag{2}$$

where $r$ is the bank's capital gain/loss; $i$ is the credit cost of banks; $\beta$ is the productivity of capital; $y(t)$ is the deviation of capital gain from its trend; $\sigma^2$ is the variance; and $\rho$ represents the negative-correlation coefficient between interest rate and capital gain[4].

Through the presented model, Stein could determine excess debt and an early warning signal of a potential crisis. As mentioned, it is this mechanism that played a role in the decreasing net worth of individuals, households, and institutions in the United States, and that was amplified by the increased leverage and pricing volatility of complex securities.

To measure the excess leveraging of banks, the introduced and defined Stein model was followed with a focus on the solution of the dynamic version of the model, which allowed for using time-series data on banks. One difference from Stein is that in this case, each bank's productivity of capital was not assumed to be deterministic or constant as in the Stein model; rather, it was calculated for the years of 2000–2018.

The optimal debt level was calculated for the years 2000 until 2018; thus, excess debt, which is the measure of overleveraging in this paper, was estimated. To calculate the banks' optimal debt ratios, data on the banks' capital gain/loss, market interest rates, and the productivity of capital were collected. Using these variables, the risk and return components of the model were then calculated. Using the above-mentioned variables, the optimal and actual debt ratios were calculated for a sample of eleven banks: four Islamic and seven conventional banks[5]. The full calculations are presented in 11 tables[6] with 18 columns each. Data were derived from the banks' annual reports, balance sheets, and financial statements, in addition to some data received directly from bank managers. The calculations are summarized below.

Column 1 consists of capital gain/loss that represents the return in percentage to the investors of the bank from capital appreciation or loss in a particular year. This capital gain/loss is calculated by dividing the change in each bank's stock-market cap by the beginning market cap at each period. The market caps were Hodrick–Prescott (HP)-filtered to eliminate the effects of daily stock-market swings. HP Filter is a data-smoothing technique frequently applied on time-series data to remove short-term fluctuations associated with the business cycle.

Column 2 represents the market interest rate. The 10-year treasury yield was used to represent the market interest rate and is therefore presented in percentage[7].

In Column 3, beta ($\beta$) represents the productivity of capital. The beta is calculated as the bank's annual gross revenue divided by total capital. The total capital here is calculated as the shareholder equity plus half of both short-term and total long-term debt[8]. To determine shareholder equity, I obtained the annual value of each bank's shareholder equity from the balance sheet. Short-term debt comprises all the banks' current liabilities that are usually due within 12 months. Long-term debts, on the other hand, are calculated as the combinations of long-term liabilities and other liabilities in banks' balance sheets. These are basically all bank liabilities due in more than a year's time. Therefore, each bank's productivity of capital is calculated for the years of 2000–2016, and it is not constant as in Stein (2012b).

Columns 4 through 9 are the risk elements in the model (Stein 2012a). Column 4 represents beta variance, which is calculated as the difference between each year's beta from the mean beta for the years 2000 and 2018, representing the deviation of each period's beta from the mean. Column 5 is also a component of the risk element. This is calculated as one-half of the square of the capital gain variable. Column 6 is the statistical correlation between interest rate and capital gain variables over the period from 2000 to 2018[9]. Columns 7 and 8 are the variance for the interest rate and capital gain variables, respectively. Each period's variance is calculated as the deviation of that period's value from the mean. Therefore, interest rate variance is the difference between each year's 10-year treasury yield and the mean interest rate from 2000 to 2018. Similarly, capital gain variance is the difference between each year's capital gain/loss and the mean capital gain from 2000 to 2018.

Column 9, which is the product of the correlation between the stochastic variables (interest rate and capital gain) and interest rate and capital gain variance, represents an additional component of the risk element. It is calculated as the product of the correlation factor of the stochastic variables (Column 6), interest rate variance (Column 7), and capital

gain variance (Column 8)[10]. Columns 10–12 are used to determine the risk that investors bear when they decide to hold equity in the bank, and this is a key issue for the investors' decision making.

Columns 10 and 11 represent the standard deviations of the interest rate and capital gains, respectively. Therefore, Column 10 is the standard deviation of values in Column 2, while Column 11 is the standard deviation of values in Column 1. Here, standard deviations are constant over the periods, as in the Stein model.

Column 12, on the other hand, is calculated as twice the value of variances of the two stochastic variables and their correlation. This is, therefore, calculated as 2 multiplied by Column 9.

Column 13 is, hence, the risk of an investor holding the equity of the bank at each time period as in Equation (A2). The risk is calculated using Columns 10–12. The risk is calculated by adding the standard deviations of the interest rate (Column 10) plus the standard deviation of capital gain (Column 11) minus the risk component in Column 12.

In the model, the optimal debt ratio maximizes the difference between net return and risk term. Therefore, only if the net return exceeds the risk premium as the optimal debt ratio become positive. The optimal debt ratio, therefore, is not a constant, as Stein also noted (Stein 2012a), but rather varies directly with net return and risk.

In Column 14, I then calculated, using all the above-mentioned variables, the Stein optimal debt ratio, $f^*(t)$. Debt ratios were normalized to remove the effects of seasonality. Therefore, normalized $f^*(t)$ measures the deviation of the optimal debt ratio away from the mean. Negative values in Column 14 represent lower optimal debt ratios away from the mean ratio during the applicable periods. The components of the optimal debt ratio are, therefore, primarily the capital gains for equity holders of the bank's stock, the market interest rate, and the risk term. The optimal debt ratio maximizes the difference between mean return and risk term. The formula for calculating optimal debt ratio using the above column numbers is: ((Column 1 − Column 2) + Column 3 − Column 4 − Column 5 + Column 9)/Column 1. This reiterates what was mentioned above that optimal debt ratio is positive only if the net return is greater than the risk premium, and this can intuitively be seen.

In Column 15, I calculated normalized optimal debt ratios using Column 14, which contains the mean and standard deviation of the optimal debt values[11]. In addition to calculating the optimal debt ratio, I calculated the banks' actual debt ratio in order to calculate the excess debt ratio. The actual debt ratio of the banks was equal to long-term debt divided by total assets, which are given in the banks' annual reports as well. Actual debt ratios are also normalized in the same way as optimal debt ratios are, and these are presented in Column 16. After optimal and actual debt ratios are calculated as discussed above, excess debt is calculated in the last columns as normalized actual minus optimal debt. The graphs of the two ratios, namely, actual and optimal debt ratios, are presented in Figure 1 below, which is followed by empirical analysis.

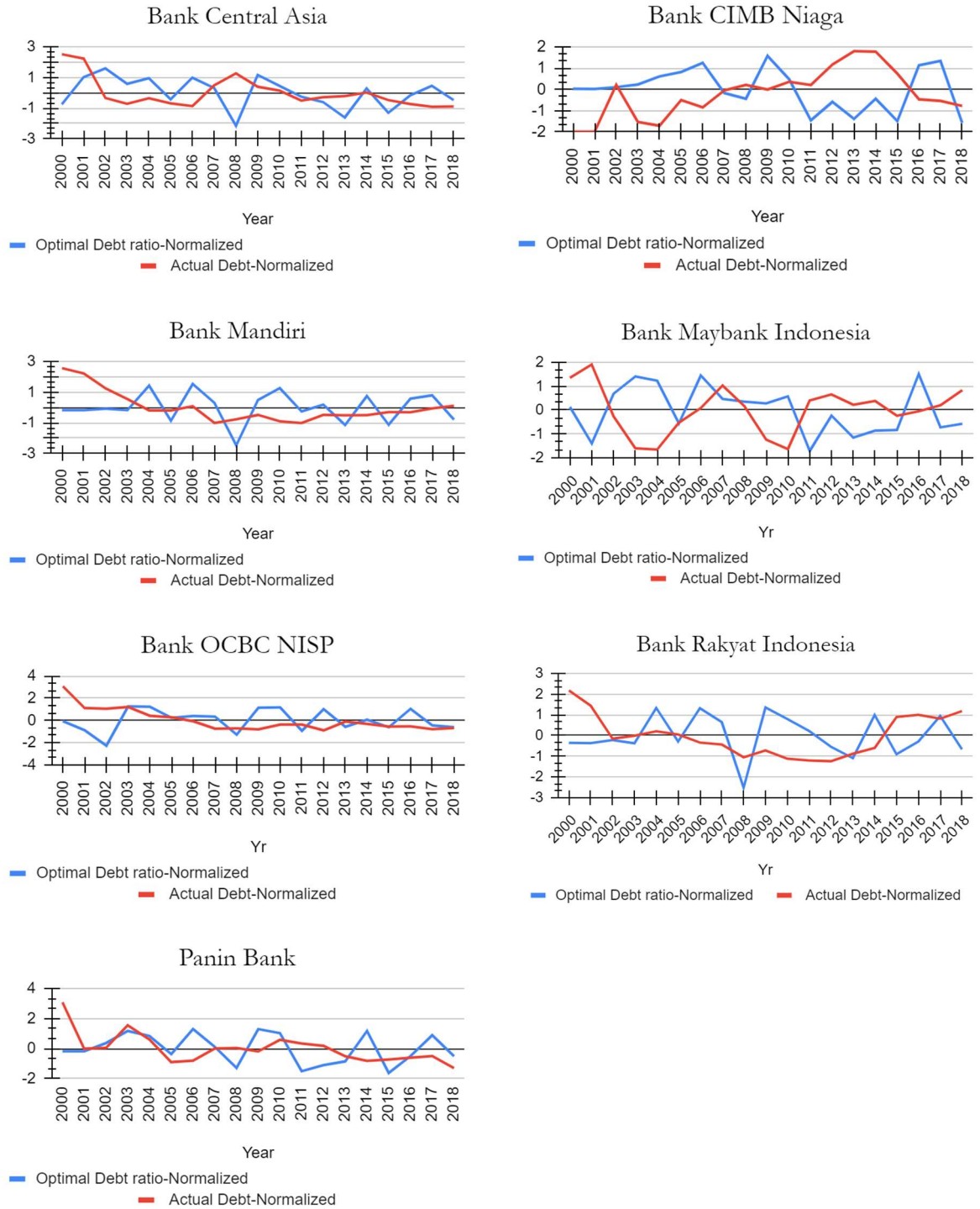

**Figure 1.** Optimal versus actual debt ratio for conventional banks. Source: author's calculations.

## 4. Empirical Analysis

### 4.1. Graphical Results and Analysis: Actual vs. Optimal Debt

Next, applying the methodology presented in the previous section, optimal leverage was calculated for a sample of eleven banks in Indonesia. For the banks under study, this analysis was performed using the banks' total long-term debts and total assets. As noted, total long-term debt represents a bank's total debt with a maturity date of more than one year from the balance sheet date. Total asset represents the value of a bank's total assets[12].

The vertical axes of Figure 1 represent the debt ratios, while horizontal axes represent the years. First, Figure 1 shows the optimal debt against actual debt ratios for the seven conventional banks under study in Indonesia. Similarly, Figure 2 shows optimal debt against the actual debt ratios for the four Islamic banks under study in Indonesia.

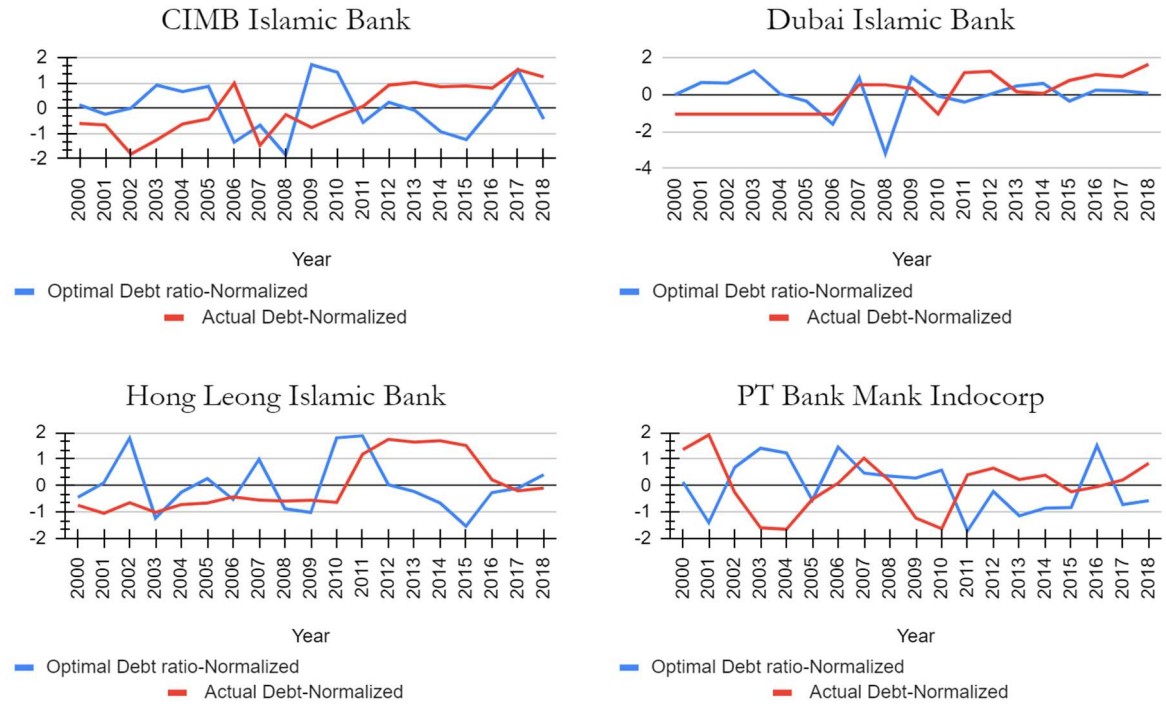

**Figure 2.** Optimal versus actual debt ratio for Islamic banks. Source: author's calculations.

The graphs show the optimal- against the actual debt ratio for the eleven banks under study for the period 2000–2018. The optimal debt ratios for most of the banks exhibited similar trends, although the trends were more pronounced for conventional than for Islamic banks. For a number of years preceding the 2007–2009 financial crisis, these banks had high optimal debt ratios. For most of the banks, about a year or two prior to 2007, optimal debt ratios began to drop, and the decrease was severe in most cases. Overall, optimal debt for most Islamic banks was not as high as optimal debt for conventional banks in this context. On the other hand, the drop in optimal debt right before the crisis for both types of banks exhibited the same behavior. Another interesting observation is that optimal debt for Islamic banks was more stable during the years immediately following the crisis when compared with that of conventional banks, with the exception of a couple of conventional banks. Moreover, the trend of actual debt exceeding optimal debt clearly reversed post-crisis for most banks in the sample.

Each of the banks' actual debt ratio was also calculated. As Figures 1 and 2 show, most of the conventional banks exhibited an increase in actual debt ratio over the years prior to the crisis, with a smoother and less severe increase in the debt ratio for the Islamic banks. More specifically, the Bank of Central Asia, Mandiri, NISP, and Panin Bank have kept the levels of optimal and actual debt ratios close. As for the rest of the traditional banks, actual debt is apart from the optimal debt, either below the optimal ratio or above it. Overall, it seems that periods of low leverage follow excess leverage periods as banks try to keep their debt below the optimal ratio to balance out the different peaks of high debt, which are way above the optimal course. In general, there are five peaks of overleveraging periods for traditional banks, whereas there are around six periods of stable debt levels over time. It is noticed that the traditional banks' ratios had gone up toward the absolute value of 3, while Islamic banks reached a maximum ratio at below the absolute value of 2. Thus, Islamic banking seems less volatile, with an actual debt level orbiting around the respective

optimal level for each bank, unlike traditional banks in many instances. Overall, we notice three periods of overleveraging that Islamic banks have experienced.

*4.2. Graphical Results and Analysis: Excess Debt*

For further analysis, the deviation of the banks' actual debt ratios from the mean over the period 2000–2018 was calculated as an alternative proxy for excess debt ratio. Excess debt ratios were calculated for the eleven above-mentioned banks. For this calculation, each bank's total assets and long-term debts were used to calculate debt ratios as previously noted. Total assets represent the bank's total balance sheet assets at the end of the period. Long-term debts are the balance of each bank's debts that are more than one year due at the end of each period. On the basis of the presented calculation method, the excess debt of each bank was calculated[13], and graphs of the excess debt ratios are exhibited in Figures 3 and 4.

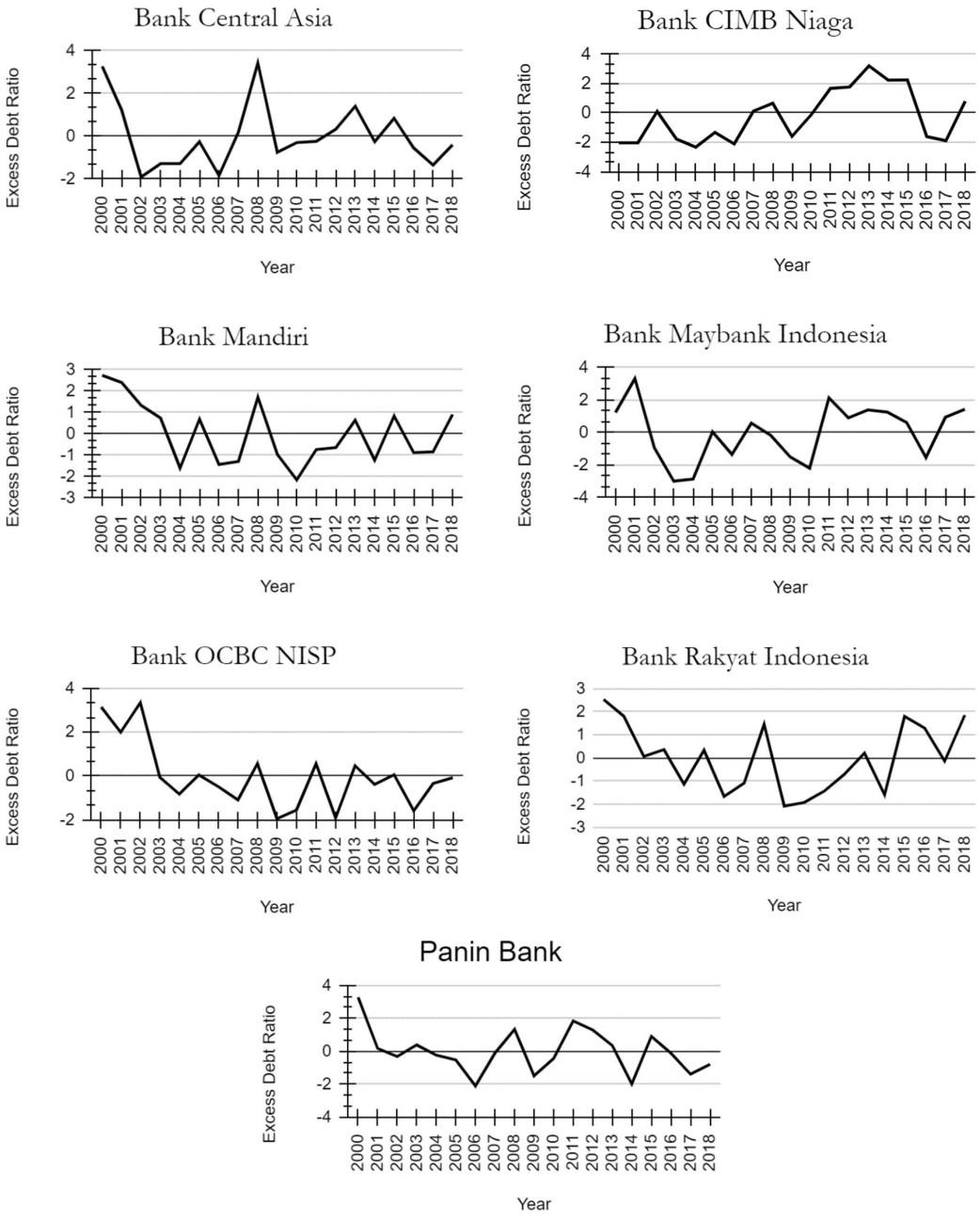

**Figure 3.** Excess debt ratios for conventional banks. Source: author's calculations.

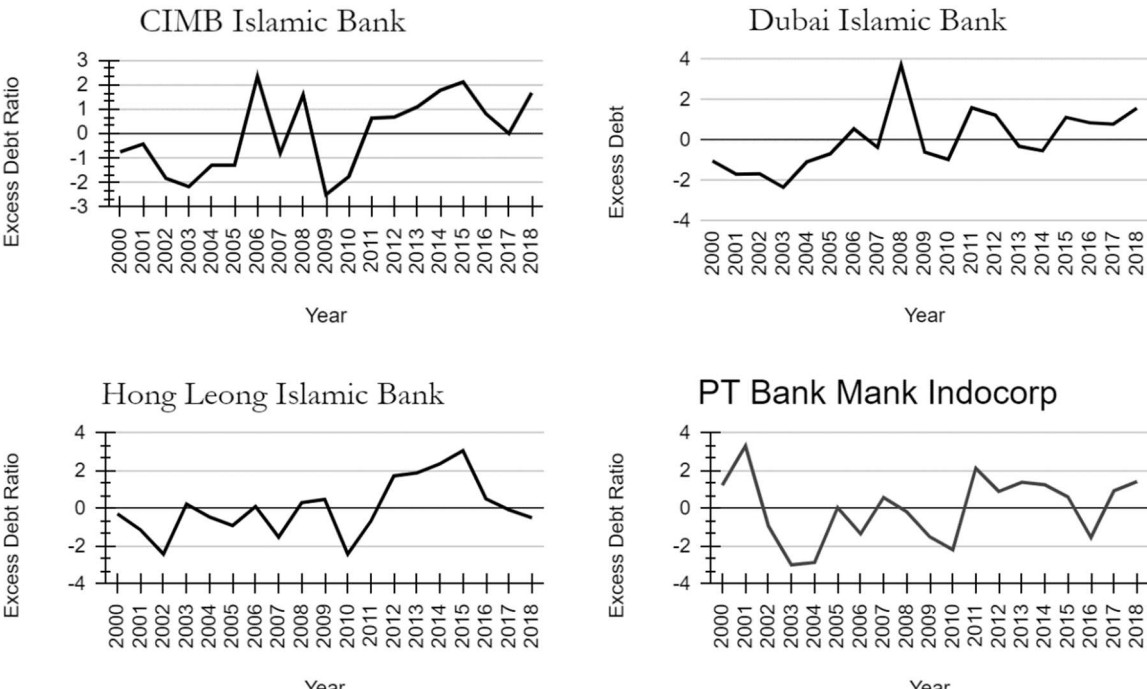

**Figure 4.** Excess debt ratio for Islamic banks. Source: author's calculations.

In the graphs in Figures 3 and 4, vertical axes represent the deviation of debt ratios, namely, excess debt ratio, and the horizontal axes represent years. Graphs show that all banks in this study exhibited a similar movement in debt ratio for most of the period with excess debt between 2007 and 2009 during the financial crisis.

For conventional banks, we observe a decline in excess debt between the years 2000 and 2003. However, this reversed and peaked from 2005 until 2008 when excess debt for all banks started to increase at a high rate. Right after the GFC, we see a sharp decline until 2010, where we can observe a rising trend once again during the period 2010–2015; however, the trend was not as steep as was observed in 2006–2008. After 2015, there was a period of recovery that continued until 2017. During this period, all conventional banks had sustainable debt levels but started to take risk and excessively leverage in 2018 and onwards. This can clearly be seen in the graphs above with the line of excess debt rising for all traditional banks.

Generally, the Islamic banks followed a similar trend, with low levels of debt from 2000 to 2003. Then, from 2004 to 2006, some banks kept a constant debt while others exhibited an increase. From 2007 to 2008, there was a rise in the excess debt but not as high as in the traditional banks, as we can see from the graphs. All banks in Indonesia, including Islamic banks, had excess debt in the years preceding and during the GFC, with excess debt higher for Islamic banks after the GFC. This occurred because Indonesian Islamic banks also highly relied on leverage, which made them vulnerable to the second-round effect of the global crisis (Hasan and Dridi 2010).

Most of the banks exhibited a rise in excess debt between 2010 and 2012, overlapping a weak economic growth due to the global economy, with the sovereign debt crisis in Europe and policy uncertainties in the United States impacting investments and companies. At times of low economic growth, companies begin to face liquidity issues and, therefore, request more loans, which increase the leverage of banks.

Table 1 and below showcases the highest excess debt for conventional and Islamic banks, respectively, during the full period under study as well as the average high excess debt, and Table 2 presents the same for the conventional banks.

**Table 1.** High Excess Leverage among Conventional Banks.

| # | Traditional Banks | High Excess Debt Year(s) | Debt Ratio | AVRG Excess Ratio |
|---|---|---|---|---|
| 1 | Bank Central Asia | 2000<br>2008 | 3.26<br>3.41 | 3.33 |
| 2 | Bank CIMB Niaga | 2013 | 2.19 | 2.19 |
| 3 | Bank Mandiri | 2000<br>2001<br>2008 | 2.73<br>2.39<br>1.70 | 2.27 |
| 4 | Bank Maybank Indonesia | 2001<br>2011 | 3.31<br>2.12 | 2.71 |
| 5 | Bank OCBC NISP | 2000<br>2002 | 3.13<br>3.33 | 3.23 |
| 6 | Bank Rakyat Indonesia | 2000<br>2001<br>2008<br>2015<br>2018 | 2.52<br>1.80<br>1.46<br>1.80<br>1.85 | 1.89 |
| 7 | Panin Bank | 2000<br>2008<br>2011 | 3.27<br>1.33<br>1.84 | 2.14 |

**Table 2.** High Excess Leverage among Islamic Banks.

| # | Islamic Banks | High Excess Debt Year(s) | Debt Ratio | AVRG Excess Ratio |
|---|---|---|---|---|
| 1 | CIMB Islamic Bank | 2006<br>2008<br>2015 | 2.33<br>1.59<br>2.12 | 2.02 |
| 2 | Dubai Islamic Bank | 2008<br>2011<br>2018 | 3.71<br>1.59<br>1.56 | 2.29 |
| 3 | Hong Leong Islamic Bank | 2012<br>2015 | 1.71<br>3.01 | 2.36 |
| 4 | PT Bank Mank Indocorp | 2001<br>2011 | 3.31<br>2.12 | 2.71 |

Table 1 presents the conventional banks that have had the highest leverage ratios. For instance, although Bank Central Asia and Bank OCBC NISP only had two years of high leverage, their ratios had been extremely high compared to the other banks. The average ratio for Central Asia and OCBC NISP was 3.33 and 3.23, respectively. On the other hand, we can see that although Bank Rakyat Indonesia had high leverage for five years—considerably more frequent high leverage compared to its peers—its average excess ratio was relatively lower.

As we can see from Table 2, the two Islamic banks, namely, CIMB Islamic Bank and Dubai Bank, had the highest leverage. The high leverage continued for three years as opposed to other banks, which only have two years of high overleverage. However, we notice that PT Bank Mank Indocorp had the highest average excess leverage ratio (2.71) among the Islamic banks.

Tables 3 and 4 exhibit the lowest debt levels based on these years for conventional and Islamic banks in our sample, respectively.

**Table 3.** Low Excess Leverage among Conventional Banks.

| # | Traditional Banks | Lowest Debt Year(s) | Debt Ratio | AVRG Debt Ratio |
|---|---|---|---|---|
| 1 | Bank Central Asia | 2002 | −1.92 | −1.61 |
| | | 2004 | −1.30 | |
| | | 2006 | −1.85 | |
| | | 2017 | −1.36 | |
| 2 | Bank CIMB Niaga | 2000 | −2.02 | −1.91 |
| | | 2001 | −2.02 | |
| | | 2003 | −1.77 | |
| | | 2004 | −2.32 | |
| | | 2006 | −2.10 | |
| | | 2009 | −1.59 | |
| | | 2016 | −1.60 | |
| | | 2017 | −1.88 | |
| 3 | Bank Mandiri | 2004 | −1.61 | −1.63 |
| | | 2006 | −1.45 | |
| | | 2007 | −1.30 | |
| | | 2010 | −2.16 | |
| 4 | Bank Maybank Indonesia | 2003 | −3.01 | −2.09 |
| | | 2004 | −2.89 | |
| | | 2006 | −1.36 | |
| | | 2009 | −1.52 | |
| | | 2010 | −2.21 | |
| | | 2016 | −1.56 | |
| 5 | Bank OCBC NISP | 2009 | −1.94 | −1.74 |
| | | 2010 | −1.55 | |
| | | 2012 | −1.90 | |
| | | 2016 | −1.58 | |
| 6 | Bank Rakyat Indonesia | 2006 | −1.66 | −1.73 |
| | | 2009 | −2.08 | |
| | | 2010 | −1.92 | |
| | | 2011 | −1.43 | |
| | | 2014 | −1.58 | |
| 7 | Panin Bank | 2006 | −2.10 | −1.86 |
| | | 2009 | −1.49 | |
| | | 2014 | −1.98 | |

**Table 4.** Low Excess Leverage among Islamic Banks.

| # | Islamic Banks | Low Debt Year(s) | Debt Ratio | AVRG Debt Ratio |
|---|---|---|---|---|
| 1 | CIMB Islamic Bank | 2003 | −2.16 | −2.16 |
| | | 2002 | −1.82 | |
| | | 2009 | −2.49 | |
| 2 | Dubai Islamic Bank | 2001 | −1.71 | −1.91 |
| | | 2002 | −1.68 | |
| | | 2003 | −2.35 | |
| 3 | Hong Leong Islamic Bank | 2002 | −2.43 | −2.14 |
| | | 2007 | −1.54 | |
| | | 2010 | −2.44 | |
| 4 | PT Bank Mank Indocorp | 2003 | −3.01 | −2.24 |
| | | 2004 | −2.89 | |
| | | 2009 | −1.52 | |
| | | 2010 | −2.21 | |
| | | 2016 | −1.56 | |

When looking at Table 3, we see that Bank CIMB Niaga had the longest time of having low leverage. For eight years, its ratio was low, below optimal debt. However, the lowest average ratio was the Bank of Maybank Indonesia with a ratio of −2.09.

From the above, we can see that the years 2000, 2001, 2011, and 2008 have the highest overleverage ratio for both traditional and Islamic banks. The worst year, 2008, had the highest overleverage with six banks having an excess ratio greater than 1.3. (See Appendix C for detailed calculations).

However, it can be noted that most Islamic banks did better than their counterparts given that they had a lower frequency of high overleverage.

On the other hand, we see the years 2009 and 2010 reflect the highest frequency of banks with low leverage, where the majority of banks were conventional. An explanation can be that after periods of high leverage before 2008, banks' sustainability to lend decreased, which forced the banks to lower their debt levels. Notice how in 2008, there were zero banks with low leverage.

As we can see from Table 4, PT Bank Mank Indocorp had the longest time of having a low leverage compared to the rest of the banks, although the bank exhibited the highest average overleveraging ratio during 2001 and 2011. We can observe how after those years of credit booms, a period of low leverage and credit crunch occurred in order to balance out the overleveraging and bring the bank to stability once again. Empirically, a long period of deleveraging nearly always follows a major financial crisis, and these episodes have been painful.

In conclusion, we clearly see that conventional banks showed high excess debt before and during the GFC, whereas Islamic banks maintained a low excess debt level during and after the crisis. This is due to their lower leverage and higher solvency, which allowed them to meet relatively stronger demand for credit; however, for both types of banks, an increase in excess debt a couple of years preceding the crisis and/or during the crisis was observed, although the level of increase differed among banks. Overall, Islamic banks leveraged less than conventional banks. To maintain high profits, conventional banks relied on loans, especially loans with high risks. This made the banks even more vulnerable due to the real estate boom. In addition, banks increased lending for corporate takeovers and leveraged buyouts, which are highly leveraged transaction loans.

## 5. Conclusions

With respect to our main endeavor, empirical analysis showed that leverage is lower in general at Islamic banks compared with their conventional counterparts in Indonesia. This study presented a model that helped identify the early warning signs of banking crises on the basis of the presence of excess debt or what is called "overleveraging". Here, a measure of overleveraging was defined as the difference of actual and sustainable debt. Furthermore, an empirical study was conducted for eleven banks in Indonesia that examined the vulnerability of these banks, and the credit and output contractions that could subsequently arise when hit by a crisis.

Results showed that all banks in the sample exhibited high excess debt levels preceding the GFC. It was argued that excess debt, rather than actual debt, could serve as an early indicator of crises because the presence of excess debt is actually the reason why banks collapse. Although banks have different debt ratios, a significant rise in excess debt ratios often precedes and/or overlaps a banking crisis. Although Islamic banks exhibited a lower level of leverage, their operations were yet characterized by a high degree of financial risk. The reason is that Islamic banks undertake risky operations in order to be able to generate a return that can attract customers when they cannot be guaranteed a return on deposits. What might make Islamic banks unique and attractive in the coming years is the short-term funding and noninterest-earning assets that might be in high need and are considered the main aspects of Islamic banks' profits. The results of this paper show that Islamic and conventional banks in Indonesia had a consistent behavior with similar banks in Bahrain, Kuwait, Malaysia, the United States, and the United Kingdom (see Issa 2020).

As seen above, the excess leverage taken by the banking sector in Indonesia poses serious threats. The unprecedented policy response to the COVID-19 pandemic has helped prevent a meltdown and maintain the flow of credit to the economy. However, the outlook remains highly uncertain, and vulnerabilities are rising, representing a potential early warning signal. The performance evaluation of Islamic banks and the stability of these types of banks are especially important today. In the aftermath of all that is currently happening after the COVID-19 crisis, we can probably anticipate a rise in non-Western conventional banking/leveraging activities. It would be interesting for future research to look into a concrete indication through data; however, it would seem logical that as conventional funds are leaving emerging markets, these economies have to rely on their specific advantages to draw additional funds—here, the focus on Islamic banking would be ratified. A better understanding of these policy questions requires specific knowledge about the riskiness of the banks and the leverage level undertaken by Islamic banks as well as their profitability.

**Funding:** This research received no external funding.

**Institutional Review Board Statement:** Not available.

**Informed Consent Statement:** Not available.

**Data Availability Statement:** Not available.

**Acknowledgments:** The author thanks Tabata Patino for data collection.

**Conflicts of Interest:** The author declares no conflict of interest.

## Appendix A

*Appendix A.1. Mathematical Derivation of Stein's Optimal Debt*

Here, Stein shows how the optimal debt ratio is derived in the logarithm case.
The stochastic differential equation is (A1)

$$dX(t) = X(t)\left[(1 + f(t))\left(\frac{dP(t)}{P(t)} + \beta(t)dt\right) - i(t)f(t) - cdt\right] \tag{A1}$$

where the debt ratio is $f(t) = \frac{L(t)}{X(t)} = \frac{Debt}{Net\ Worth}$; capital gain or loss is $\frac{dP(t)}{P(t)}$; productivity of capital is $\beta(t) = \frac{Income}{Assets}$; interest rate is $i(t)$; $(1 + f(t)) = \frac{Assets}{Net\ Worth}$; and ratio of consumption is: $c = \frac{(Consumption\ or\ Dividends)}{Net\ Worth}$ and is taken as given.

Let the price evolve as:

$$dP(t) = P(t)\big(\alpha(t)dt + \sigma_p dw_p(t)\big), \tag{A2}$$

where $\alpha(t)$ represents the asset's drift component and the interest rate is represented by the sum of $i$ and a Brownian Motion term as follows:

$$i(t) = idt + \sigma_i dw_i(t). \tag{A3}$$

Substitute (A2) and (A3) in (A1) and derive (A4):

$$dX(t) = X(t)\big[(1 + f(t))\big((\alpha(t)dt + \sigma_p dw_p(t)) + \beta(t)dt\big) - i(t)f(t)dt - cdt\big]\, dX(t) = X(t)(Mf(t))dt + X(t)\beta f(t) \tag{A4}$$

$$Mf(t) = [(1 + f(t))((\alpha(t)dt) + \beta(t)dt) - i(t)f(t)dt - cdt]$$
$$\beta(t) = [(1 + f(t))\sigma_p dw_p - \sigma_i f(t)dw_i(t)]$$
$$\beta^2 f(t) = [(1 + f(t)^2)\sigma^2_p\, dt + f(t)^2\sigma^2_i\, dt - 2f(t)(1 + f(t))\sigma_i\sigma_p dw_p dw_i]$$
$$\text{Risk} = Rf(t) = \left(\frac{1}{2}\right)\big[(1 + f(t)^2)\sigma^2_p\, dt + f(t)^2\sigma^2_i\, dt - 2f(t)(1 + f(t))\sigma_i\sigma_p\big].$$

$Mf(t)$ contains the deterministic terms and $\beta(t)$ contains the stochastic terms. To solve for $X(t)$, consider the change in $lnX(t)$ in (A5). This is based upon the Ito equation of the

stochastic calculus. A great virtue of using the logarithm criterion is that one does not need to use dynamic programming. The expectation of $dlnX(t)$ is (A6):

$$dlnX(t) = \left(\frac{1}{X(t)}\right)dXt - (\frac{1}{2}X(t)^2)(dx(t)^2) \tag{A5}$$

$$E[d(lnX(t))] = [Mf(t)] - R[f(t)]dt. \tag{A6}$$

The correlation $\rho dt = E(dw_p dw_i)$ is negative, which increases risk: $(dt)^2 = 0, dwdt = 0$.

The optimal debt ratio $f^*$ maximizes the difference between the mean and risk:

$$f^* = argmax_f[M(f(t)) - R(f(t))] = [\alpha(t) + \beta(t) - i] - [\left(\frac{(\sigma_p^2 - \rho\sigma_i\sigma_p)}{\sigma^2}\right)]$$
$$f^* = argmax_f[M(f(t)) - R(f(t))] = f^*(t) = \{(r - i) + \beta - \alpha y(t) - (\frac{1/2(\sigma_p^2 - \rho_{ip}\sigma_i\sigma_p)}{\sigma^2})\} \tag{A7}$$

s.t.

$$Risk = \sigma^2 = \sigma_i{}^2 + \sigma_p{}^2 - (2\rho_{ip}\sigma_i\sigma_p). $$

Appendix A.1.1. Model I

Model I assumes that the price $P(t)$ has a trend $rt$ and a deviation $Y(t)$ from it (A8). The deviation $Y(t)$ follows an Ornstein–Uhlenbeck ergodic mean reverting process (A9). Coefficient $\alpha$ is positive and finite. The interest rate is the same as in model II:

$$P(t) = Pexp(rt + y(t)). \tag{A8}$$

The deviation from the trend is demonstrated through:

$$y(t) = ln\ P(t) - ln\ P - rt.$$

The mean reversion aspect characterized by a convergence of $\alpha$ is defined as:

$$dy(t) = -\alpha(t)dt + \sigma_p dw_p(t). \tag{A9}$$

In this model, Stein defines $E(dw) = 0; E(dw)^2 = dt$:

$$limy(t) \sim N\left(0, \frac{\sigma^2}{2\alpha}\right).$$

Stein constrains the solution such that $r \leq i$ and calls this the "No free lunch constraint". Therefore, using the stochastic calculus in model I is the first term in (A10):

$$dP(t) = P(t)(\alpha(t)dt + \sigma_p dw_p(t)) \quad dP(t)/P(t) = (r - \alpha y(t)) + \frac{1}{2}\sigma_p{}^2)dt + \sigma_p dw_p, \tag{A10}$$

where $\alpha(t)$ represents the asset's drift component, and the interest rate is represented by the sum of $i$ and a Brownian Motion term as follows:

$$i(t) = idt + \sigma_i dw_i(t).$$

Substituting (A10) in (A7) and deriving (A11), the optimal debt ratio in model I is as follows:

$$f^*(t) = \left[(r - i) + \beta - \alpha y(t) - \frac{\left(\frac{1}{2}\right)\left(\sigma_p^2 - \sigma_i\sigma_p\rho\right)}{\sigma^2}\right]. \tag{A11}$$

Consider $\beta(t)$ as deterministic.

Appendix A.1.2. Model II

In model II, the price equation is (A12). The drift is $\alpha(t)dt = \boldsymbol{\pi}\mathrm{dt}$, and the diffusion is $\sigma_p dw_p$ :

$$dP(t)/P(t) = \boldsymbol{\pi}\mathrm{dt} + \sigma_p dw_p \tag{A12}$$

The optimal debt ratio $f^*(t)$ is (A13). Consider $\beta(t)$ as deterministic:

$$f^*(t) = [(\boldsymbol{\pi} + \beta(t) - i) - \frac{\left(\sigma_p^2 - \sigma_i \sigma_p \rho\right)}{\sigma^2}] \tag{A13}$$

s.t.

$$\sigma^2 = {\sigma_i}^2 + {\sigma_p}^2 - \left(2\rho_{ip}\sigma_i\sigma_p\right).$$

In terms of a maximization portfolio decision, we have:

$$\max_{\alpha_t}[\alpha_t(E(R_{t+1}) - R_{F, t+1}) - \frac{k}{2}\alpha_t^2\sigma_t^2]. \tag{A14}$$

## Appendix B

**Table A1.** List of Banks.

| Bank Name | Type | Country |
|---|---|---|
| Central Asia Bank | Conventional Bank | Indonesia |
| CIMB Bank Niaga | Conventional Bank | Indonesia |
| Mandiri Bank | Conventional Bank | Indonesia |
| Maybank Indonesia Bank | Conventional Bank | Indonesia |
| OCBC NISP Bank | Conventional Bank | Indonesia |
| Bank Rakyat Indonesia | Conventional Bank | Indonesia |
| Panin Bank | Conventional Bank | Indonesia |
| CIMB Bank | Islamic Bank | Indonesia |
| Dubai Islamic Bank | Islamic Bank | Indonesia |
| Hong Leong Bank | Islamic Bank | Indonesia |
| PT Mank Indocorp Bank | Islamic Bank | Indonesia |

## Appendix C

**Table A2.** Calculations of Optimal and Excess Debt: CIMB Islamic Bank.

| Bank | Year | Capital Gains/(Losses), (r) | Interest Rate (i) | Beta (Productivity of Capital, β) | Beta Variance (αγ(t)) | Half Square of Capital Gain Variance | Correlation of Interest and Capital Gain Variables | Interest Rate Variance | Capital Gain Variance | Correlation and Variances of Interest and Capital Gain | Std. Deviation of Interest Rate | Std. Deviation of Capital Gain | 2 × (Correlation and Variances of Interest and Capital Gain) | Risk | Optimal Debt Ratio, f*(t) | Normalized Optimal Debt Ratio | Actual Debt Ratio | Normalized Actual Debt Ratio | Excess Debt |
|---|---|---|---|---|---|---|---|---|---|---|---|---|---|---|---|---|---|---|---|
| CIMB Islamic Bank | 2000 | 0.0096 | 0.0258 | 0.3368 | (0.020) | 0.000 | (0.36) | (0.0098) | 0.3260 | 0.0011 | 0.0133 | 0.6503 | 0.0023 | 0.661 | 0.52 | 0.13 | 0.0494 | (0.61) | (0.74) |
| | 2001 | −0.0965 | 0.0243 | 0.3065 | (0.051) | 0.005 | (0.36) | (0.0113) | 0.0636 | 0.0003 | 0.0133 | 0.6503 | 0.0005 | 0.663 | 0.35 | (0.25) | 0.0485 | (0.67) | (0.42) |
| | 2002 | −0.0424 | 0.0209 | 0.2710 | (0.086) | 0.001 | (0.36) | (0.0147) | 1.2205 | 0.0064 | 0.0133 | 0.6503 | 0.0127 | 0.651 | 0.46 | (0.00) | 0.0296 | (1.82) | (1.82) |
| | 2003 | 0.2783 | 0.0188 | 0.3168 | (0.040) | 0.039 | (0.36) | (0.0168) | −0.1798 | (0.0011) | 0.0133 | 0.6503 | (0.0021) | 0.666 | 0.87 | 0.91 | 0.0388 | (1.26) | (2.16) |
| | 2004 | 0.1899 | 0.0286 | 0.3015 | (0.056) | 0.018 | (0.36) | (0.0070) | −0.0292 | (0.0001) | 0.0133 | 0.6503 | (0.0001) | 0.664 | 0.75 | 0.66 | 0.0491 | (0.63) | (1.28) |
| | 2005 | 0.2486 | 0.0191 | 0.3377 | (0.020) | 0.031 | (0.36) | (0.0165) | 0.2895 | 0.0017 | 0.0133 | 0.6503 | 0.0034 | 0.660 | 0.84 | 0.86 | 0.0524 | (0.43) | (1.28) |
| | 2006 | 2.3655 | 0.0197 | 0.3522 | (0.005) | 2.798 | (0.36) | (0.0159) | −0.2307 | (0.0013) | 0.0133 | 0.6503 | (0.0026) | 0.666 | (0.14) | (1.35) | 0.0755 | 0.99 | 2.33 |
| | 2007 | −0.1993 | 0.0339 | 0.4041 | 0.047 | 0.020 | (0.36) | (0.0017) | −0.6142 | (0.0004) | 0.0133 | 0.6503 | (0.0007) | 0.664 | 0.16 | (0.68) | 0.0352 | (1.47) | (0.80) |
| | 2008 | −0.4600 | 0.0373 | 0.4205 | 0.063 | 0.106 | (0.26) | 0.0017 | 0.8440 | (0.0005) | 0.0133 | 0.6503 | (0.0010) | 0.665 | (0.37) | (1.85) | 0.0551 | (0.26) | 1.59 |
| | 2009 | 1.1797 | 0.0252 | 0.4292 | 0.072 | 0.696 | (0.36) | (0.0104) | 0.0327 | 0.0001 | 0.0133 | 0.6503 | 0.0002 | 0.663 | 1.23 | 1.72 | 0.0468 | (0.77) | (2.49) |
| | 2010 | 0.5658 | 0.0374 | 0.4022 | 0.045 | 0.160 | (0.36) | 0.0018 | −0.5646 | 0.0004 | 0.0133 | 0.6503 | 0.0007 | 0.663 | 1.10 | 1.42 | 0.0540 | (0.33) | (1.74) |
| | 2011 | −0.1541 | 0.0476 | 0.4411 | 0.084 | 0.012 | (0.36) | 0.0120 | 1.1362 | (0.0049) | 0.0133 | 0.6503 | (0.0097) | 0.673 | 0.21 | (0.57) | 0.0605 | 0.07 | 0.64 |
| | 2012 | 0.0637 | 0.0442 | 0.3922 | 0.035 | 0.002 | (0.36) | 0.0086 | 0.3179 | (0.0010) | 0.0133 | 0.6503 | (0.0020) | 0.665 | 0.56 | 0.23 | 0.0743 | 0.91 | 0.68 |
| | 2013 | −0.0338 | 0.0422 | 0.3907 | 0.033 | 0.001 | (0.36) | 0.0066 | 0.1408 | (0.0003) | 0.0133 | 0.6503 | (0.0007) | 0.664 | 0.42 | (0.09) | 0.0760 | 1.01 | 1.10 |
| | 2014 | −0.2548 | 0.0415 | 0.3313 | (0.026) | 0.032 | (0.36) | 0.0059 | −0.0714 | 0.0002 | 0.0133 | 0.6503 | 0.0003 | 0.663 | 0.04 | (0.93) | 0.0732 | 0.84 | 1.77 |
| | 2015 | −0.3276 | 0.0405 | 0.3714 | 0.014 | 0.054 | (0.36) | 0.0049 | 0.2717 | (0.0005) | 0.0133 | 0.6503 | (0.0010) | 0.664 | (0.10) | (1.25) | 0.0737 | 0.87 | 2.12 |
| | 2016 | −0.0089 | 0.0504 | 0.3416 | (0.016) | 0.000 | (0.36) | 0.0148 | −0.1599 | 0.0008 | 0.0133 | 0.6503 | 0.0017 | 0.662 | 0.45 | (0.02) | 0.0724 | 0.79 | 0.81 |
| | 2017 | 0.6666 | 0.0516 | 0.3112 | (0.046) | 0.222 | (0.36) | 0.0160 | −0.1599 | 0.0009 | 0.0133 | 0.6503 | 0.0018 | 0.662 | 1.13 | 1.51 | 0.0843 | 1.52 | 0.01 |
| | 2018 | −0.1107 | 0.0666 | 0.3293 | (0.028) | 0.006 | (0.36) | 0.0310 | −0.1599 | 0.0018 | 0.0133 | 0.6503 | 0.0035 | 0.660 | 0.27 | (0.43) | 0.0796 | 1.24 | 1.67 |

**Table A3.** Calculations of Optimal and Excess Debt: Dubai Islamic Bank.

| Bank | Year | Capital Gains/(Losses), (r) | Interest Rate (i) | Beta (Productivity of Capital, β) | Beta Variance (αγ(t)) | Half Square of Capital Gain Variance | Correlation of Interest and Capital Gain Variables | Interest Rate Variance | Capital Gain Variance | Correlation and Variances of Interest and Capital Gain | Std. Deviation of Interest Rate | Std. Deviation of Capital Gain | 2 × (Correlation and Variances of Interest and Capital Gain) | Risk | Optimal Debt Ratio, f*(t) | Normalized Optimal Debt Ratio | Actual Debt Ratio | Normalized Actual Debt Ratio | Excess Debt |
|---|---|---|---|---|---|---|---|---|---|---|---|---|---|---|---|---|---|---|---|
| Dubai Islamic Bank | 2000 | 0.0000 | 0.0258 | 0.6658 | 0.278 | 0.000 | (0.27) | (0.0098) | 0.3260 | 0.0009 | 0.0133 | 0.7859 | 0.0017 | 0.797 | 0.46 | (0.02) | 0.0000 | (1.06) | (1.04) |
| | 2001 | 0.2875 | 0.0243 | 0.6203 | 0.232 | 0.041 | (0.27) | (0.0113) | 0.0636 | 0.0002 | 0.0133 | 0.7859 | 0.0004 | 0.799 | 0.76 | 0.65 | 0.0000 | (1.06) | (1.71) |
| | 2002 | 0.2524 | 0.0209 | 0.5176 | 0.130 | 0.032 | (0.27) | (0.0147) | 1.2205 | 0.0048 | 0.0133 | 0.7859 | 0.0097 | 0.790 | 0.75 | 0.62 | 0.0000 | (1.06) | (1.68) |
| | 2003 | 0.8120 | 0.0188 | 0.5501 | 0.162 | 0.330 | (0.27) | (0.0168) | −0.1798 | (0.0008) | 0.0133 | 0.7859 | (0.0016) | 0.801 | 1.06 | 1.29 | 0.0000 | (1.06) | (2.35) |
| | 2004 | 1.9775 | 0.0286 | 0.3991 | 0.011 | 1.955 | (0.27) | (0.0070) | −0.0292 | (0.0001) | 0.0133 | 0.7859 | (0.0001) | 0.799 | 0.48 | 0.03 | 0.0000 | (1.06) | (1.09) |
| | 2005 | 2.1303 | 0.0191 | 0.4578 | 0.070 | 2.269 | (0.27) | (0.0165) | 0.2895 | 0.0013 | 0.0133 | 0.7859 | 0.0026 | 0.797 | 0.29 | (0.37) | 0.0000 | (1.06) | (0.69) |
| | 2006 | −0.4808 | 0.0197 | 0.3583 | (0.030) | 0.116 | (0.27) | (0.0159) | −0.2307 | (0.0010) | 0.0133 | 0.7859 | (0.0020) | 0.801 | (0.29) | (1.61) | 0.0000 | (1.06) | 0.55 |
| | 2007 | 0.4568 | 0.0339 | 0.5096 | 0.122 | 0.104 | (0.27) | (0.0017) | −0.6142 | (0.0003) | 0.0133 | 0.7859 | (0.0006) | 0.800 | 0.88 | 0.90 | 0.0327 | 0.53 | (0.37) |
| | 2008 | −0.8265 | 0.0373 | 0.4183 | 0.030 | 0.342 | (0.27) | 0.0017 | 0.8440 | (0.0004) | 0.0133 | 0.7859 | (0.0008) | 0.800 | (1.02) | (3.19) | 0.0325 | 0.52 | 3.71 |
| | 2009 | 0.4674 | 0.0252 | 0.4615 | 0.073 | 0.109 | (0.27) | (0.0104) | 0.0327 | 0.0001 | 0.0133 | 0.7859 | 0.0002 | 0.799 | 0.90 | 0.94 | 0.0286 | 0.34 | (0.61) |
| | 2010 | −0.0137 | 0.0374 | 0.2758 | (0.112) | 0.000 | (0.27) | 0.0018 | −0.5646 | 0.0003 | 0.0133 | 0.7859 | 0.0006 | 0.799 | 0.42 | (0.09) | 0.0000 | (1.06) | (0.97) |
| | 2011 | −0.1101 | 0.0476 | 0.3097 | (0.078) | 0.006 | (0.27) | 0.0120 | 1.1362 | (0.0037) | 0.0133 | 0.7859 | (0.0074) | 0.807 | 0.27 | (0.41) | 0.0461 | 1.18 | 1.59 |
| | 2012 | 0.0361 | 0.0442 | 0.2719 | (0.116) | 0.001 | (0.27) | 0.0086 | 0.3179 | (0.0007) | 0.0133 | 0.7859 | (0.0015) | 0.801 | 0.47 | 0.02 | 0.0474 | 1.25 | 1.23 |
| | 2013 | 1.7768 | 0.0422 | 0.2734 | (0.115) | 1.578 | (0.27) | 0.0066 | 0.1408 | (0.0003) | 0.0133 | 0.7859 | (0.0005) | 0.800 | 0.68 | 0.47 | 0.0248 | 0.15 | (0.32) |
| | 2014 | 0.2873 | 0.0415 | 0.2954 | (0.093) | 0.041 | (0.27) | 0.0059 | −0.0714 | 0.0001 | 0.0133 | 0.7859 | 0.0002 | 0.799 | 0.74 | 0.60 | 0.0230 | 0.06 | (0.54) |
| | 2015 | −0.1043 | 0.0405 | 0.2601 | (0.128) | 0.005 | (0.27) | 0.0049 | 0.2717 | (0.0004) | 0.0133 | 0.7859 | (0.0007) | 0.800 | 0.30 | (0.36) | 0.0374 | 0.76 | 1.12 |
| | 2016 | 0.1265 | 0.0504 | 0.2329 | (0.155) | 0.008 | (0.27) | 0.0148 | −0.1599 | 0.0006 | 0.0133 | 0.7859 | 0.0013 | 0.798 | 0.57 | 0.24 | 0.0440 | 1.08 | 0.85 |
| | 2017 | 0.1113 | 0.0516 | 0.2479 | (0.140) | 0.006 | (0.27) | 0.0160 | −0.1599 | 0.0007 | 0.0133 | 0.7859 | 0.0014 | 0.798 | 0.55 | 0.20 | 0.0418 | 0.97 | 0.78 |
| | 2018 | 0.0749 | 0.0666 | 0.2471 | (0.141) | 0.003 | (0.27) | 0.0310 | −0.1599 | 0.0013 | 0.0133 | 0.7859 | 0.0027 | 0.797 | 0.50 | 0.07 | 0.0553 | 1.63 | 1.56 |

**Table A4.** Calculations of Optimal and Excess Debt: Hong Leong Islamic Bank.

| Bank | Year | Capital Gains/(Losses), (r) | Interest Rate (i) | Beta (Productivity of Capital, β) | Beta Variance (αy(t)) | Half Square of Capital Gain Variance | Correlation of Interest and Capital Gain Variables | Interest Rate Variance | Capital Gain Variance | Correlation and Variances of Interest and Capital Gain | Std. Deviation of Interest Rate | Std. Deviation of Capital Gain | 2 × (Correlation and Variances of Interest and Capital Gain) | Risk | Optimal Debt Ratio, f*(t) | Normalized Optimal Debt Ratio | Actual Debt Ratio | Normalized Actual Debt Ratio | Excess Debt |
|---|---|---|---|---|---|---|---|---|---|---|---|---|---|---|---|---|---|---|---|
| Hong Leong Islamic Bank | 2000 | 0.0096 | 0.0258 | 0.5835 | 0.216 | 0.000 | 0.15 | (0.0098) | 0.3260 | (0.0005) | 0.0133 | 0.2561 | (0.0010) | 0.270 | 1.30 | (0.45) | 0.0057 | (0.75) | (0.30) |
| | 2001 | 0.1144 | 0.0243 | 0.4867 | 0.119 | 0.007 | 0.15 | (0.0113) | 0.0636 | (0.0001) | 0.0133 | 0.2561 | (0.0002) | 0.270 | 1.67 | 0.09 | 0.0000 | (1.05) | (1.15) |
| | 2002 | 0.6370 | 0.0209 | 0.4504 | 0.083 | 0.203 | 0.15 | (0.0147) | 1.2205 | (0.0027) | 0.0133 | 0.2561 | (0.0054) | 0.275 | 2.83 | 1.78 | 0.0075 | (0.65) | (2.43) |
| | 2003 | −0.1364 | 0.0188 | 0.4679 | 0.100 | 0.009 | 0.15 | (0.0168) | −0.1798 | 0.0005 | 0.0133 | 0.2561 | 0.0009 | 0.268 | 0.76 | (1.24) | 0.0007 | (1.02) | 0.22 |
| | 2004 | 0.0476 | 0.0286 | 0.3300 | (0.038) | 0.001 | 0.15 | (0.0070) | −0.0292 | 0.0000 | 0.0133 | 0.2561 | 0.0001 | 0.269 | 1.43 | (0.26) | 0.0062 | (0.73) | (0.47) |
| | 2005 | 0.1459 | 0.0191 | 0.2556 | (0.112) | 0.011 | 0.15 | (0.0165) | 0.2895 | (0.0007) | 0.0133 | 0.2561 | (0.0014) | 0.271 | 1.78 | 0.26 | 0.0072 | (0.67) | (0.92) |
| | 2006 | −0.0139 | 0.0197 | 0.3151 | (0.052) | 0.000 | 0.15 | (0.0159) | −0.2307 | 0.0006 | 0.0133 | 0.2561 | 0.0011 | 0.268 | 1.25 | (0.53) | 0.0116 | (0.43) | 0.09 |
| | 2007 | 0.3373 | 0.0339 | 0.4206 | 0.053 | 0.057 | 0.15 | (0.0017) | −0.6142 | 0.0002 | 0.0133 | 0.2561 | 0.0003 | 0.269 | 2.28 | 0.98 | 0.0094 | (0.55) | (1.54) |
| | 2008 | −0.0595 | 0.0373 | 0.4176 | 0.050 | 0.002 | 0.15 | 0.0017 | 0.8440 | 0.0002 | 0.0133 | 0.2561 | 0.0004 | 0.269 | 1.00 | (0.88) | 0.0087 | (0.59) | 0.29 |
| | 2009 | −0.0942 | 0.0252 | 0.5064 | 0.139 | 0.004 | 0.15 | (0.0104) | 0.0327 | (0.0001) | 0.0133 | 0.2561 | (0.0001) | 0.270 | 0.90 | (1.03) | 0.0092 | (0.56) | 0.46 |
| | 2010 | 0.6427 | 0.0374 | 0.3539 | (0.014) | 0.207 | 0.15 | 0.0018 | −0.5646 | (0.0002) | 0.0133 | 0.2561 | (0.0003) | 0.270 | 2.84 | 1.80 | 0.0077 | (0.64) | (2.44) |
| | 2011 | 0.6704 | 0.0476 | 0.2554 | (0.112) | 0.225 | 0.15 | 0.0120 | 1.1362 | 0.0021 | 0.0133 | 0.2561 | 0.0042 | 0.265 | 2.89 | 1.87 | 0.0418 | 1.18 | (0.70) |
| | 2012 | 0.1200 | 0.0442 | 0.3313 | (0.036) | 0.007 | 0.15 | 0.0086 | 0.3179 | 0.0004 | 0.0133 | 0.2561 | 0.0008 | 0.269 | 1.63 | 0.03 | 0.0523 | 1.74 | 1.71 |
| | 2013 | 0.0662 | 0.0422 | 0.3155 | (0.052) | 0.002 | 0.15 | 0.0066 | 0.1408 | 0.0001 | 0.0133 | 0.2561 | 0.0003 | 0.269 | 1.45 | (0.23) | 0.0504 | 1.64 | 1.87 |
| | 2014 | −0.0156 | 0.0415 | 0.3095 | (0.058) | 0.000 | 0.15 | 0.0059 | −0.0714 | (0.0001) | 0.0133 | 0.2561 | (0.0001) | 0.270 | 1.15 | (0.67) | 0.0514 | 1.69 | 2.35 |
| | 2015 | −0.1664 | 0.0405 | 0.3262 | (0.041) | 0.014 | 0.15 | 0.0049 | 0.2717 | 0.0002 | 0.0133 | 0.2561 | 0.0004 | 0.269 | 0.55 | (1.55) | 0.0481 | 1.51 | 3.06 |
| | 2016 | 0.0688 | 0.0504 | 0.2860 | (0.082) | 0.002 | 0.15 | 0.0148 | −0.1599 | (0.0004) | 0.0133 | 0.2561 | (0.0007) | 0.270 | 1.42 | (0.28) | 0.0238 | 0.22 | 0.49 |
| | 2017 | 0.1036 | 0.0516 | 0.2923 | (0.075) | 0.005 | 0.15 | 0.0160 | −0.1599 | (0.0004) | 0.0133 | 0.2561 | (0.0008) | 0.270 | 1.53 | (0.11) | 0.0160 | (0.20) | (0.09) |
| | 2018 | 0.2391 | 0.0666 | 0.2804 | (0.087) | 0.029 | 0.15 | 0.0310 | −0.1599 | (0.0008) | 0.0133 | 0.2561 | (0.0015) | 0.271 | 1.89 | 0.40 | 0.0178 | (0.10) | (0.51) |

**Table A5.** Calculations of Optimal and Excess Debt: PT Bank Mank Indocorp.

| Bank | Year | Capital Gains/(Losses), (r) | Interest Rate (i) | Beta (Productivity of Capital, β) | Beta Variance (αy(t)) | Half Square of Capital Gain Variance | Correlation of Interest and Capital Gain Variables | Interest Rate Variance | Capital Gain Variance | Correlation and Variances of Interest and Capital Gain | Std. Deviation of Interest Rate | Std. Deviation of Capital Gain | 2 × (Correlation and Variances of Interest and Capital Gain) | Risk | Optimal Debt Ratio, f*(t) | Normalized Optimal Debt Ratio | Actual Debt Ratio | Normalized Actual Debt Ratio | Excess Debt |
|---|---|---|---|---|---|---|---|---|---|---|---|---|---|---|---|---|---|---|---|
| PT Bank Mank Indocorp | 2000 | 0.0096 | 0.0258 | 1.0100 | (0.003) | 0.000 | (0.23) | (0.0098) | 0.3260 | 0.0007 | 0.0133 | 0.6297 | 0.0015 | 0.641 | 1.55 | 0.13 | 0.1110 | 1.35 | 1.22 |
| | 2001 | −0.4186 | 0.0243 | 5.3172 | 4.304 | 0.088 | (0.23) | (0.0113) | 0.0636 | 0.0002 | 0.0133 | 0.6297 | 0.0003 | 0.643 | 0.75 | (1.40) | 0.1292 | 1.91 | 3.31 |
| | 2002 | 0.1956 | 0.0209 | 0.8513 | (0.162) | 0.019 | (0.23) | (0.0147) | 1.2205 | 0.0041 | 0.0133 | 0.6297 | 0.0083 | 0.635 | 1.85 | 0.68 | 0.0581 | (0.27) | (0.95) |
| | 2003 | 1.3398 | 0.0188 | 0.9427 | (0.070) | 0.898 | (0.23) | (0.0168) | −0.1798 | (0.0007) | 0.0133 | 0.6297 | (0.0014) | 0.644 | 2.23 | 1.41 | 0.0144 | (1.60) | (3.01) |
| | 2004 | 0.5260 | 0.0286 | 0.9078 | (0.105) | 0.138 | (0.23) | (0.0070) | −0.0292 | (0.0000) | 0.0133 | 0.6297 | (0.0001) | 0.643 | 2.13 | 1.23 | 0.0126 | (1.66) | (2.89) |
| | 2005 | −0.2082 | 0.0191 | 0.7268 | (0.286) | 0.022 | (0.23) | (0.0165) | 0.2895 | 0.0011 | 0.0133 | 0.6297 | 0.0022 | 0.641 | 1.19 | (0.56) | 0.0495 | (0.53) | 0.03 |
| | 2006 | 0.7075 | 0.0197 | 0.7680 | (0.245) | 0.250 | (0.23) | (0.0159) | −0.2307 | (0.0008) | 0.0133 | 0.6297 | (0.0017) | 0.645 | 2.25 | 1.45 | 0.0696 | 0.09 | (1.36) |
| | 2007 | 0.1469 | 0.0339 | 0.6602 | (0.353) | 0.011 | (0.23) | (0.0017) | −0.6142 | (0.0002) | 0.0133 | 0.6297 | (0.0005) | 0.643 | 1.73 | 0.47 | 0.1006 | 1.03 | 0.56 |
| | 2008 | 0.1070 | 0.0373 | 1.0879 | 0.075 | 0.006 | (0.23) | 0.0017 | 0.8440 | (0.0003) | 0.0133 | 0.6297 | (0.0007) | 0.644 | 1.67 | 0.35 | 0.0721 | 0.16 | (0.19) |
| | 2009 | 0.0655 | 0.0252 | 0.8851 | (0.128) | 0.002 | (0.23) | (0.0104) | 0.0327 | 0.0001 | 0.0133 | 0.6297 | 0.0002 | 0.643 | 1.64 | 0.28 | 0.0264 | (1.24) | (1.52) |
| | 2010 | 1.8075 | 0.0374 | 0.9296 | (0.083) | 1.633 | (0.23) | 0.0018 | −0.5646 | 0.0002 | 0.0133 | 0.6297 | 0.0005 | 0.642 | 1.79 | 0.57 | 0.0133 | (1.64) | (2.21) |
| | 2011 | −0.4712 | 0.0476 | 0.7533 | (0.260) | 0.111 | (0.23) | 0.0120 | 1.1362 | (0.0032) | 0.0133 | 0.6297 | (0.0063) | 0.649 | 0.59 | (1.72) | 0.0799 | 0.40 | 2.12 |
| | 2012 | −0.0850 | 0.0442 | 0.7047 | (0.308) | 0.004 | (0.23) | 0.0086 | 0.3179 | (0.0006) | 0.0133 | 0.6297 | (0.0013) | 0.644 | 1.37 | (0.23) | 0.0882 | 0.65 | 0.89 |
| | 2013 | −0.3435 | 0.0422 | 0.8080 | (0.205) | 0.059 | (0.23) | 0.0066 | 0.1408 | (0.0002) | 0.0133 | 0.6297 | (0.0004) | 0.643 | 0.88 | (1.15) | 0.0741 | 0.22 | 1.37 |
| | 2014 | −0.2701 | 0.0415 | 0.6546 | (0.358) | 0.036 | (0.23) | 0.0059 | −0.0714 | 0.0001 | 0.0133 | 0.6297 | 0.0002 | 0.643 | 1.03 | (0.86) | 0.0794 | 0.38 | 1.25 |
| | 2015 | −0.2625 | 0.0405 | 0.6685 | (0.344) | 0.034 | (0.23) | 0.0049 | 0.2717 | (0.0003) | 0.0133 | 0.6297 | (0.0006) | 0.644 | 1.05 | (0.84) | 0.0590 | (0.24) | 0.60 |
| | 2016 | 1.0347 | 0.0504 | 0.5866 | (0.426) | 0.535 | (0.23) | 0.0148 | −0.1599 | 0.0005 | 0.0133 | 0.6297 | 0.0011 | 0.642 | 2.28 | 1.50 | 0.0651 | (0.05) | (1.56) |
| | 2017 | −0.2263 | 0.0516 | 0.5460 | (0.467) | 0.026 | (0.23) | 0.0160 | −0.1599 | 0.0006 | 0.0133 | 0.6297 | 0.0012 | 0.642 | 1.11 | (0.73) | 0.0735 | 0.20 | 0.93 |
| | 2018 | −0.1732 | 0.0666 | 0.4354 | (0.577) | 0.015 | (0.23) | 0.0310 | −0.1599 | 0.0011 | 0.0133 | 0.6297 | 0.0023 | 0.641 | 1.19 | (0.58) | 0.0941 | 0.83 | 1.41 |

**Table A6.** Calculations of Optimal and Excess Debt: Bank Central Asia.

| Bank | Year | Capital Gains/(Losses), (r) | Interest Rate (i) | Beta (Productivity of Capital, β) | Beta Variance (αy(t)) | Half Square of Capital Gain Variance | Correlation of Interest and Capital Gain Variables | Interest Rate Variance | Capital Gain Variance | Correlation and Variances of Interest and Capital Gain | Std. Deviation of Interest Rate | Std. Deviation of Capital Gain | 2 × (Correlation and Variances of Interest and Capital Gain) | Risk | Optimal debt Ratio, f*(t) | Normalized Optimal Debt Ratio | Actual Debt Ratio | Normalized Actual Debt Ratio | Excess Debt |
|---|---|---|---|---|---|---|---|---|---|---|---|---|---|---|---|---|---|---|---|
| Bank Central Asia | 2000 | 0.0096 | 0.0258 | 1.6925 | 0.812 | 0.000 | (0.46) | (0.0098) | 0.3260 | 0.0015 | 0.0133 | 0.3366 | 0.0029 | 0.347 | 2.50 | (0.75) | 0.0157 | 2.50 | 3.26 |
| | 2001 | 0.6546 | 0.0243 | 1.3646 | 0.484 | 0.214 | (0.46) | (0.0113) | 0.0636 | 0.0003 | 0.0133 | 0.3366 | 0.0007 | 0.349 | 3.71 | 1.02 | 0.0144 | 2.23 | 1.21 |
| | 2002 | 0.9932 | 0.0209 | 1.2234 | 0.343 | 0.493 | (0.46) | (0.0147) | 1.2205 | 0.0082 | 0.0133 | 0.3366 | 0.0165 | 0.333 | 4.10 | 1.58 | 0.0033 | (0.34) | (1.92) |
| | 2003 | 0.4399 | 0.0188 | 1.0072 | 0.126 | 0.097 | (0.46) | (0.0168) | −0.1798 | (0.0014) | 0.0133 | 0.3366 | (0.0028) | 0.353 | 3.41 | 0.58 | 0.0016 | (0.72) | (1.30) |
| | 2004 | 0.6291 | 0.0286 | 0.9449 | 0.064 | 0.198 | (0.46) | (0.0070) | −0.0292 | (0.0001) | 0.0133 | 0.3366 | (0.0002) | 0.350 | 3.67 | 0.95 | 0.0032 | (0.35) | (1.30) |
| | 2005 | 0.0796 | 0.0191 | 0.9418 | 0.061 | 0.003 | (0.46) | (0.0165) | 0.2895 | 0.0022 | 0.0133 | 0.3366 | 0.0044 | 0.346 | 2.72 | (0.43) | 0.0018 | (0.69) | (0.27) |
| | 2006 | 0.6682 | 0.0197 | 0.9890 | 0.108 | 0.223 | (0.46) | (0.0159) | −0.2307 | (0.0017) | 0.0133 | 0.3366 | (0.0034) | 0.353 | 3.69 | 0.99 | 0.0010 | (0.87) | (1.85) |
| | 2007 | 0.3442 | 0.0339 | 0.8848 | 0.004 | 0.059 | (0.46) | (0.0017) | −0.6142 | (0.0005) | 0.0133 | 0.3366 | (0.0009) | 0.351 | 3.22 | 0.31 | 0.0069 | 0.48 | 0.17 |
| | 2008 | −0.2675 | 0.0373 | 1.0585 | 0.178 | 0.036 | (0.46) | 0.0017 | 0.8440 | (0.0007) | 0.0133 | 0.3366 | (0.0014) | 0.351 | 1.54 | (2.15) | 0.0103 | 1.26 | 3.41 |
| | 2009 | 0.7827 | 0.0252 | 0.8477 | (0.033) | 0.306 | (0.46) | (0.0104) | 0.0327 | 0.0002 | 0.0133 | 0.3366 | 0.0003 | 0.350 | 3.81 | 1.16 | 0.0065 | 0.39 | (0.77) |
| | 2010 | 0.3932 | 0.0374 | 0.7625 | (0.118) | 0.077 | (0.46) | 0.0018 | −0.5646 | 0.0005 | 0.0133 | 0.3366 | 0.0010 | 0.349 | 3.32 | 0.45 | 0.0054 | 0.14 | (0.31) |
| | 2011 | 0.2276 | 0.0476 | 0.7361 | (0.145) | 0.026 | (0.46) | 0.0120 | 1.1362 | (0.0063) | 0.0133 | 0.3366 | (0.0126) | 0.363 | 2.84 | (0.26) | 0.0026 | (0.51) | (0.25) |
| | 2012 | 0.0834 | 0.0442 | 0.6646 | (0.216) | 0.003 | (0.46) | 0.0086 | 0.3179 | (0.0013) | 0.0133 | 0.3366 | (0.0025) | 0.352 | 2.60 | (0.61) | 0.0035 | (0.30) | 0.31 |
| | 2013 | −0.1580 | 0.0422 | 0.7218 | (0.159) | 0.012 | (0.46) | 0.0066 | 0.1408 | (0.0004) | 0.0133 | 0.3366 | (0.0009) | 0.351 | 1.90 | (1.61) | 0.0038 | (0.22) | 1.39 |
| | 2014 | 0.3386 | 0.0415 | 0.6927 | (0.188) | 0.057 | (0.46) | 0.0059 | −0.0714 | 0.0002 | 0.0133 | 0.3366 | 0.0004 | 0.350 | 3.21 | 0.28 | 0.0048 | 0.01 | (0.27) |
| | 2015 | −0.0910 | 0.0405 | 0.6506 | (0.230) | 0.004 | (0.46) | 0.0049 | 0.2717 | (0.0006) | 0.0133 | 0.3366 | (0.0012) | 0.351 | 2.12 | (1.30) | 0.0027 | (0.47) | 0.83 |
| | 2016 | 0.1926 | 0.0504 | 0.5543 | (0.326) | 0.019 | (0.46) | 0.0148 | −0.1599 | 0.0011 | 0.0133 | 0.3366 | 0.0022 | 0.348 | 2.89 | (0.18) | 0.0016 | (0.73) | (0.56) |
| | 2017 | 0.4079 | 0.0516 | 0.5127 | (0.368) | 0.083 | (0.46) | 0.0160 | −0.1599 | 0.0012 | 0.0133 | 0.3366 | 0.0024 | 0.348 | 3.32 | 0.45 | 0.0008 | (0.91) | (1.36) |
| | 2018 | 0.1182 | 0.0666 | 0.4842 | (0.397) | 0.007 | (0.46) | 0.0310 | −0.1599 | 0.0023 | 0.0133 | 0.3366 | 0.0046 | 0.345 | 2.69 | (0.48) | 0.0009 | (0.89) | (0.42) |

**Table A7.** Calculations of Optimal and Excess Debt: Bank CIMB Niaga.

| Bank | Year | Capital Gains/(Losses), (r) | Interest Rate (i) | Beta (Productivity of Capital, β) | Beta Variance (αy(t)) | Half Square of Capital Gain Variance | Correlation of Interest and Capital Gain Variables | Interest Rate Variance | Capital Gain Variance | Correlation and Variances of Interest and Capital Gain | Std. Deviation of Interest Rate | Std. Deviation of Capital Gain | 2 × (Correlation and Variances of Interest and Capital Gain) | Risk | Optimal Debt Ratio, f*(t) | Normalized Optimal Debt Ratio | Actual Debt Ratio | Normalized Actual Debt Ratio | Excess Debt |
|---|---|---|---|---|---|---|---|---|---|---|---|---|---|---|---|---|---|---|---|
| Bank CIMB Niaga | 2000 | 0.0000 | 0.0258 | 0.0000 | (0.698) | 0.000 | (0.28) | (0.0098) | 0.3260 | 0.0009 | 0.0133 | 0.6074 | 0.0018 | 0.619 | 1.09 | 0.02 | 0.0000 | (1.58) | (1.60) |
| | 2001 | 0.0000 | 0.0243 | 0.0000 | (0.698) | 0.000 | (0.28) | (0.0113) | 0.0636 | 0.0002 | 0.0133 | 0.6074 | 0.0004 | 0.620 | 1.09 | 0.02 | 0.0000 | (1.58) | (1.60) |
| | 2002 | 0.0000 | 0.0209 | 0.9650 | 0.267 | 0.000 | (0.28) | (0.0147) | 1.2205 | 0.0050 | 0.0133 | 0.6074 | 0.0099 | 0.611 | 1.12 | 0.08 | 0.0380 | 0.37 | 0.29 |
| | 2003 | 0.0626 | 0.0188 | 0.9524 | 0.254 | 0.002 | (0.28) | (0.0168) | −0.1798 | (0.0008) | 0.0133 | 0.6074 | (0.0017) | 0.622 | 1.19 | 0.23 | 0.0079 | (1.17) | (1.40) |
| | 2004 | 0.1977 | 0.0286 | 0.8405 | 0.142 | 0.020 | (0.28) | (0.0070) | −0.0292 | (0.0001) | 0.0133 | 0.6074 | (0.0001) | 0.621 | 1.37 | 0.61 | 0.0048 | (1.33) | (1.94) |
| | 2005 | 0.2559 | 0.0191 | 0.8527 | 0.155 | 0.033 | (0.28) | (0.0165) | 0.2895 | 0.0013 | 0.0133 | 0.6074 | 0.0026 | 0.618 | 1.46 | 0.81 | 0.0256 | (0.26) | (1.08) |
| | 2006 | 1.5233 | 0.0197 | 1.0814 | 0.383 | 1.160 | (0.28) | (0.0159) | −0.2307 | (0.0010) | 0.0133 | 0.6074 | (0.0020) | 0.623 | 1.67 | 1.26 | 0.0198 | (0.56) | (1.82) |
| | 2007 | −0.0432 | 0.0339 | 0.8396 | 0.141 | 0.001 | (0.28) | (0.0017) | −0.6142 | (0.0003) | 0.0133 | 0.6074 | (0.0006) | 0.621 | 1.00 | (0.17) | 0.0336 | 0.14 | 0.31 |
| | 2008 | −0.1140 | 0.0373 | 1.0852 | 0.387 | 0.006 | (0.28) | 0.0017 | 0.8440 | (0.0004) | 0.0133 | 0.6074 | (0.0008) | 0.621 | 0.87 | (0.44) | 0.0380 | 0.37 | 0.82 |
| | 2009 | 0.7135 | 0.0252 | 0.8379 | 0.140 | 0.255 | (0.28) | (0.0104) | 0.0327 | 0.0001 | 0.0133 | 0.6074 | 0.0002 | 0.620 | 1.82 | 1.58 | 0.0342 | 0.18 | (1.40) |
| | 2010 | 1.8402 | 0.0374 | 0.7871 | 0.089 | 1.693 | (0.28) | 0.0018 | −0.5646 | 0.0003 | 0.0133 | 0.6074 | 0.0006 | 0.620 | 1.30 | 0.48 | 0.0405 | 0.50 | 0.02 |
| | 2011 | −0.3413 | 0.0476 | 0.7497 | 0.052 | 0.058 | (0.28) | 0.0120 | 1.1362 | (0.0038) | 0.0133 | 0.6074 | (0.0076) | 0.628 | 0.39 | (1.45) | 0.0379 | 0.37 | 1.82 |
| | 2012 | −0.1444 | 0.0442 | 0.6455 | (0.053) | 0.010 | (0.28) | 0.0086 | 0.3179 | (0.0008) | 0.0133 | 0.6074 | (0.0015) | 0.622 | 0.80 | (0.59) | 0.0544 | 1.22 | 1.80 |
| | 2013 | −0.3379 | 0.0422 | 0.6866 | (0.012) | 0.057 | (0.28) | 0.0066 | 0.1408 | (0.0003) | 0.0133 | 0.6074 | (0.0005) | 0.621 | 0.42 | (1.39) | 0.0653 | 1.77 | 3.17 |
| | 2014 | −0.1114 | 0.0415 | 0.6405 | (0.058) | 0.006 | (0.28) | 0.0059 | −0.0714 | 0.0001 | 0.0133 | 0.6074 | 0.0002 | 0.620 | 0.87 | (0.44) | 0.0649 | 1.75 | 2.19 |
| | 2015 | −0.3608 | 0.0405 | 0.6620 | (0.036) | 0.065 | (0.28) | 0.0049 | 0.2717 | (0.0004) | 0.0133 | 0.6074 | (0.0007) | 0.621 | 0.37 | (1.49) | 0.0471 | 0.84 | 2.33 |
| | 2016 | 0.4533 | 0.0504 | 0.5870 | (0.111) | 0.103 | (0.28) | 0.0148 | −0.1599 | 0.0007 | 0.0133 | 0.6074 | 0.0013 | 0.619 | 1.61 | 1.13 | 0.0263 | (0.23) | (1.36) |
| | 2017 | 0.5797 | 0.0516 | 0.5155 | (0.183) | 0.168 | (0.28) | 0.0160 | −0.1599 | 0.0007 | 0.0133 | 0.6074 | 0.0014 | 0.619 | 1.71 | 1.34 | 0.0250 | (0.29) | (1.63) |
| | 2018 | −0.3617 | 0.0666 | 0.5357 | (0.162) | 0.065 | (0.28) | 0.0310 | −0.1599 | 0.0014 | 0.0133 | 0.6074 | 0.0027 | 0.618 | 0.33 | (1.58) | 0.0209 | (0.50) | 1.07 |



**Table A8.** Calculations of Optimal and Excess Debt: Bank Mandiri.

| Bank | Year | Capital Gains/(Losses), (r) | Interest Rate (i) | Beta (Productivity of Capital, β) | Beta Variance (αy(t)) | Half Square of Capital Gain Variance | Correlation of Interest and Capital Gain Variables | Interest Rate Variance | Capital Gain Variance | Correlation and Variances of Interest and Capital Gain | Std. Deviation of Interest Rate | Std. Deviation of Capital Gain | 2 × (Correlation and Variances of Interest and Capital Gain) | Risk | Optimal Debt Ratio, f*(t) | Normalized Optimal Debt Ratio | Actual Debt Ratio | Normalized Actual Debt Ratio | Excess Debt |
|---|---|---|---|---|---|---|---|---|---|---|---|---|---|---|---|---|---|---|---|
| Bank Mandiri | 2000 | 0.0000 | 0.0258 | 0.8687 | 0.137 | 0.000 | (0.23) | (0.0098) | 0.3260 | 0.0007 | 0.0133 | 0.5262 | 0.0015 | 0.538 | 1.31 | (0.18) | 0.1103 | 2.56 | 2.73 |
| | 2001 | 0.0000 | 0.0243 | 1.1538 | 0.423 | 0.000 | (0.23) | (0.0113) | 0.0636 | 0.0002 | 0.0133 | 0.5262 | 0.0003 | 0.539 | 1.31 | (0.18) | 0.1008 | 2.21 | 2.39 |
| | 2002 | 0.0000 | 0.0209 | 1.1206 | 0.389 | 0.000 | (0.23) | (0.0147) | 1.2205 | 0.0041 | 0.0133 | 0.5262 | 0.0083 | 0.531 | 1.35 | (0.12) | 0.0741 | 1.24 | 1.36 |
| | 2003 | 0.0000 | 0.0188 | 0.7271 | (0.004) | 0.000 | (0.23) | (0.0168) | −0.1798 | (0.0007) | 0.0133 | 0.5262 | (0.0014) | 0.541 | 1.32 | (0.17) | 0.0550 | 0.55 | 0.72 |
| | 2004 | 0.7583 | 0.0286 | 0.6037 | (0.128) | 0.287 | (0.23) | (0.0070) | −0.0292 | (0.0000) | 0.0133 | 0.5262 | (0.0001) | 0.540 | 2.17 | 1.42 | 0.0347 | (0.19) | (1.61) |
| | 2005 | −0.1913 | 0.0191 | 0.7214 | (0.010) | 0.018 | (0.23) | (0.0165) | 0.2895 | 0.0011 | 0.0133 | 0.5262 | 0.0022 | 0.537 | 0.94 | (0.87) | 0.0347 | (0.19) | 0.68 |
| | 2006 | 0.9704 | 0.0197 | 0.7885 | 0.057 | 0.471 | (0.23) | (0.0159) | −0.2307 | (0.0008) | 0.0133 | 0.5262 | (0.0017) | 0.541 | 2.24 | 1.54 | 0.0422 | 0.08 | (1.46) |
| | 2007 | 0.1623 | 0.0339 | 0.6997 | (0.032) | 0.013 | (0.23) | (0.0017) | −0.6142 | (0.0002) | 0.0133 | 0.5262 | (0.0005) | 0.540 | 1.57 | 0.29 | 0.0122 | (1.01) | (1.30) |
| | 2008 | −0.5165 | 0.0373 | 0.9185 | 0.187 | 0.133 | (0.23) | 0.0017 | 0.8440 | (0.0003) | 0.0133 | 0.5262 | (0.0007) | 0.540 | 0.08 | (2.46) | 0.0188 | (0.77) | 1.69 |
| | 2009 | 1.7812 | 0.0252 | 0.7547 | 0.023 | 1.586 | (0.23) | (0.0104) | 0.0327 | 0.0001 | 0.0133 | 0.5262 | 0.0002 | 0.539 | 1.67 | 0.49 | 0.0266 | (0.49) | (0.97) |
| | 2010 | 0.6247 | 0.0374 | 0.7911 | 0.060 | 0.195 | (0.23) | 0.0018 | −0.5646 | 0.0002 | 0.0133 | 0.5262 | 0.0005 | 0.539 | 2.08 | 1.26 | 0.0151 | (0.90) | (2.16) |
| | 2011 | 0.0198 | 0.0476 | 0.6436 | (0.088) | 0.000 | (0.23) | 0.0120 | 1.1362 | (0.0032) | 0.0133 | 0.5262 | (0.0063) | 0.546 | 1.28 | (0.23) | 0.0122 | (1.01) | (0.78) |
| | 2012 | 0.1387 | 0.0442 | 0.6062 | (0.125) | 0.010 | (0.23) | 0.0086 | 0.3179 | (0.0006) | 0.0133 | 0.5262 | (0.0013) | 0.541 | 1.51 | 0.19 | 0.0267 | (0.48) | (0.67) |
| | 2013 | −0.2327 | 0.0422 | 0.7117 | (0.020) | 0.027 | (0.23) | 0.0066 | 0.1408 | (0.0002) | 0.0133 | 0.5262 | (0.0004) | 0.540 | 0.79 | (1.14) | 0.0258 | (0.51) | 0.63 |
| | 2014 | 0.3439 | 0.0415 | 0.6352 | (0.096) | 0.059 | (0.23) | 0.0059 | −0.0714 | 0.0001 | 0.0133 | 0.5262 | 0.0002 | 0.539 | 1.81 | 0.74 | 0.0262 | (0.50) | (1.24) |
| | 2015 | −0.2299 | 0.0405 | 0.6381 | (0.093) | 0.026 | (0.23) | 0.0049 | 0.2717 | (0.0003) | 0.0133 | 0.5262 | (0.0006) | 0.540 | 0.80 | (1.12) | 0.0316 | (0.30) | 0.82 |
| | 2016 | 0.2806 | 0.0504 | 0.5404 | (0.191) | 0.039 | (0.23) | 0.0148 | −0.1599 | 0.0005 | 0.0133 | 0.5262 | 0.0011 | 0.538 | 1.71 | 0.57 | 0.0314 | (0.31) | (0.88) |
| | 2017 | 0.3774 | 0.0516 | 0.5007 | (0.231) | 0.071 | (0.23) | 0.0160 | −0.1599 | 0.0006 | 0.0133 | 0.5262 | 0.0012 | 0.538 | 1.83 | 0.79 | 0.0382 | (0.06) | (0.85) |
| | 2018 | −0.1317 | 0.0666 | 0.4707 | (0.261) | 0.009 | (0.23) | 0.0310 | −0.1599 | 0.0011 | 0.0133 | 0.5262 | 0.0023 | 0.537 | 0.98 | (0.80) | 0.0428 | 0.10 | 0.90 |

**Table A9.** Calculations of Optimal and Excess Debt: Maybank Indonesia.

| Bank | Year | Capital Gains/(Losses), (r) | Interest Rate (i) | Beta (Productivity of Capital, β) | Beta Variance (αy(t)) | Half Square of Capital Gain Variance | Correlation of Interest and Capital Gain Variables | Interest Rate Variance | Capital Gain Variance | Correlation and Variances of Interest and Capital Gain | Std. Deviation of Interest Rate | Std. Deviation of Capital Gain | 2 × (Correlation and Variances of Interest and Capital Gain) | Risk | Optimal Debt Ratio, f*(t) | Normalized Optimal Debt Ratio | Actual Debt Ratio | Normalized Actual Debt Ratio | Excess Debt |
|---|---|---|---|---|---|---|---|---|---|---|---|---|---|---|---|---|---|---|---|
| Maybank Indonesia | 2000 | 0.0096 | 0.0258 | 1.0100 | (0.003) | 0.000 | (0.23) | (0.0098) | 0.3260 | 0.0007 | 0.0133 | 0.6297 | 0.0015 | 0.641 | 1.55 | 0.13 | 0.1110 | 1.35 | 1.22 |
| | 2001 | −0.4186 | 0.0243 | 5.3172 | 4.304 | 0.088 | (0.23) | (0.0113) | 0.0636 | 0.0002 | 0.0133 | 0.6297 | 0.0003 | 0.643 | 0.75 | (1.40) | 0.1292 | 1.91 | 3.31 |
| | 2002 | 0.1956 | 0.0209 | 0.8513 | (0.162) | 0.019 | (0.23) | (0.0147) | 1.2205 | 0.0041 | 0.0133 | 0.6297 | 0.0083 | 0.635 | 1.85 | 0.68 | 0.0581 | (0.27) | (0.95) |
| | 2003 | 1.3398 | 0.0188 | 0.9427 | (0.070) | 0.898 | (0.23) | (0.0168) | −0.1798 | (0.0007) | 0.0133 | 0.6297 | (0.0014) | 0.644 | 2.23 | 1.41 | 0.0144 | (1.60) | (3.01) |
| | 2004 | 0.5260 | 0.0286 | 0.9078 | (0.105) | 0.138 | (0.23) | (0.0070) | −0.0292 | (0.0000) | 0.0133 | 0.6297 | (0.0001) | 0.643 | 2.13 | 1.23 | 0.0126 | (1.66) | (2.89) |
| | 2005 | −0.2082 | 0.0191 | 0.7268 | (0.286) | 0.022 | (0.23) | (0.0165) | 0.2895 | 0.0011 | 0.0133 | 0.6297 | 0.0022 | 0.641 | 1.19 | (0.56) | 0.0495 | (0.53) | 0.03 |
| | 2006 | 0.7075 | 0.0197 | 0.7680 | (0.245) | 0.250 | (0.23) | (0.0159) | −0.2307 | (0.0008) | 0.0133 | 0.6297 | (0.0017) | 0.645 | 2.25 | 1.45 | 0.0696 | 0.09 | (1.36) |
| | 2007 | 0.1469 | 0.0339 | 0.6602 | (0.353) | 0.011 | (0.23) | (0.0017) | −0.6142 | (0.0002) | 0.0133 | 0.6297 | (0.0005) | 0.643 | 1.73 | 0.47 | 0.1006 | 1.03 | 0.56 |
| | 2008 | 0.1070 | 0.0373 | 1.0879 | 0.075 | 0.006 | (0.23) | 0.0017 | 0.8440 | (0.0003) | 0.0133 | 0.6297 | (0.0007) | 0.644 | 1.67 | 0.35 | 0.0721 | 0.16 | (0.19) |
| | 2009 | 0.0655 | 0.0252 | 0.8851 | (0.128) | 0.002 | (0.23) | (0.0104) | 0.0327 | 0.0001 | 0.0133 | 0.6297 | 0.0002 | 0.643 | 1.64 | 0.28 | 0.0264 | (1.24) | (1.52) |
| | 2010 | 1.8075 | 0.0374 | 0.9296 | (0.083) | 1.633 | (0.23) | 0.0018 | −0.5646 | 0.0002 | 0.0133 | 0.6297 | 0.0005 | 0.642 | 1.79 | 0.57 | 0.0133 | (1.64) | (2.21) |
| | 2011 | −0.4712 | 0.0476 | 0.7533 | (0.260) | 0.111 | (0.23) | 0.0120 | 1.1362 | (0.0032) | 0.0133 | 0.6297 | (0.0063) | 0.649 | 0.59 | (1.72) | 0.0799 | 0.40 | 2.12 |
| | 2012 | −0.0850 | 0.0442 | 0.7047 | (0.308) | 0.004 | (0.23) | 0.0086 | 0.3179 | (0.0006) | 0.0133 | 0.6297 | (0.0013) | 0.644 | 1.37 | (0.23) | 0.0882 | 0.65 | 0.89 |
| | 2013 | −0.3435 | 0.0422 | 0.8080 | (0.205) | 0.059 | (0.23) | 0.0066 | 0.1408 | (0.0002) | 0.0133 | 0.6297 | (0.0004) | 0.643 | 0.88 | (1.15) | 0.0741 | 0.22 | 1.37 |
| | 2014 | −0.2701 | 0.0415 | 0.6546 | (0.358) | 0.036 | (0.23) | 0.0059 | −0.0714 | 0.0001 | 0.0133 | 0.6297 | 0.0002 | 0.643 | 1.03 | (0.86) | 0.0794 | 0.38 | 1.25 |
| | 2015 | −0.2625 | 0.0405 | 0.6685 | (0.344) | 0.034 | (0.23) | 0.0049 | 0.2717 | (0.0003) | 0.0133 | 0.6297 | (0.0006) | 0.644 | 1.05 | (0.84) | 0.0590 | (0.24) | 0.60 |
| | 2016 | 1.0347 | 0.0504 | 0.5866 | (0.426) | 0.535 | (0.23) | 0.0148 | −0.1599 | 0.0005 | 0.0133 | 0.6297 | 0.0011 | 0.642 | 2.28 | 1.50 | 0.0651 | (0.05) | (1.56) |
| | 2017 | −0.2263 | 0.0516 | 0.5460 | (0.467) | 0.026 | (0.23) | 0.0160 | −0.1599 | 0.0006 | 0.0133 | 0.6297 | 0.0012 | 0.642 | 1.11 | (0.73) | 0.0735 | 0.20 | 0.93 |
| | 2018 | −0.1732 | 0.0666 | 0.4354 | (0.577) | 0.015 | (0.23) | 0.0310 | −0.1599 | 0.0011 | 0.0133 | 0.6297 | 0.0023 | 0.641 | 1.19 | (0.58) | 0.0941 | 0.83 | 1.41 |

**Table A10.** Calculations of Optimal and Excess Debt: Bank OCBC NISP.

| Bank | Year | Capital Gains/(losses), (r) | Interest Rate (i) | Beta (Productivity of Capital, β) | Beta Variance (αy(t)) | Half Square of Capital Gain Variance | Correlation of Interest and Capital Gain Variables | Interest Rate Variance | Capital Gain Variance | Correlation and Variances of Interest and Capital Gain | Std. Deviation of Interest Rate | Std. Deviation of Capital Gain | 2 × (Correlation and Variances of Interest and Capital Gain) | Risk | Optimal Debt Ratio, f*(t) | Normalized Optimal Debt Ratio | Actual Debt Ratio | Normalized Actual Debt Ratio | Excess Debt |
|---|---|---|---|---|---|---|---|---|---|---|---|---|---|---|---|---|---|---|---|
| Bank OCBC NISP | 2000 | 0.0096 | 0.0258 | 0.8277 | 0.184 | 0.000 | (0.39) | (0.0098) | 0.3260 | 0.0013 | 0.0133 | 0.7017 | 0.0025 | 0.712 | 0.88 | (0.08) | 0.1509 | 3.05 | 3.13 |
| | 2001 | −0.2543 | 0.0243 | 0.8328 | 0.189 | 0.032 | (0.39) | (0.0113) | 0.0636 | 0.0003 | 0.0133 | 0.7017 | 0.0006 | 0.714 | 0.47 | (0.89) | 0.0819 | 1.10 | 1.99 |
| | 2002 | 2.6166 | 0.0209 | 0.7102 | 0.066 | 3.423 | (0.39) | (0.0147) | 1.2205 | 0.0070 | 0.0133 | 0.7017 | 0.0141 | 0.701 | (0.25) | (2.29) | 0.0799 | 1.04 | 3.33 |
| | 2003 | 1.0175 | 0.0188 | 0.8278 | 0.184 | 0.518 | (0.39) | (0.0168) | −0.1798 | (0.0012) | 0.0133 | 0.7017 | (0.0024) | 0.717 | 1.57 | 1.25 | 0.0853 | 1.20 | (0.05) |
| | 2004 | 0.9266 | 0.0286 | 0.6341 | (0.010) | 0.429 | (0.39) | (0.0070) | −0.0292 | (0.0001) | 0.0133 | 0.7017 | (0.0002) | 0.715 | 1.56 | 1.23 | 0.0575 | 0.41 | (0.82) |
| | 2005 | 0.1191 | 0.0191 | 0.7660 | 0.122 | 0.007 | (0.39) | (0.0165) | 0.2895 | 0.0019 | 0.0133 | 0.7017 | 0.0038 | 0.711 | 1.04 | 0.22 | 0.0528 | 0.28 | 0.06 |
| | 2006 | 0.2077 | 0.0197 | 0.8284 | 0.185 | 0.022 | (0.39) | (0.0159) | −0.2307 | (0.0014) | 0.0133 | 0.7017 | (0.0029) | 0.718 | 1.13 | 0.39 | 0.0399 | (0.09) | (0.48) |
| | 2007 | 0.1943 | 0.0339 | 0.6703 | 0.027 | 0.019 | (0.39) | (0.0017) | −0.6142 | (0.0004) | 0.0133 | 0.7017 | (0.0008) | 0.716 | 1.10 | 0.33 | 0.0166 | (0.75) | (1.08) |
| | 2008 | −0.3549 | 0.0373 | 0.8653 | 0.222 | 0.063 | (0.39) | 0.0017 | 0.8440 | (0.0006) | 0.0133 | 0.7017 | (0.0012) | 0.716 | 0.26 | (1.29) | 0.0175 | (0.72) | 0.57 |
| | 2009 | 0.7066 | 0.0252 | 0.6880 | 0.044 | 0.250 | (0.39) | (0.0104) | 0.0327 | 0.0001 | 0.0133 | 0.7017 | 0.0003 | 0.715 | 1.50 | 1.13 | 0.0145 | (0.81) | (1.94) |
| | 2010 | 0.7949 | 0.0374 | 0.5805 | (0.063) | 0.316 | (0.39) | 0.0018 | −0.5646 | 0.0004 | 0.0133 | 0.7017 | 0.0008 | 0.714 | 1.52 | 1.16 | 0.0294 | (0.39) | (1.55) |
| | 2011 | −0.2444 | 0.0476 | 0.6191 | (0.025) | 0.030 | (0.39) | 0.0120 | 1.1362 | (0.0054) | 0.0133 | 0.7017 | (0.0108) | 0.726 | 0.44 | (0.95) | 0.0295 | (0.38) | 0.57 |
| | 2012 | 0.6320 | 0.0442 | 0.5040 | (0.140) | 0.200 | (0.39) | 0.0086 | 0.3179 | (0.0011) | 0.0133 | 0.7017 | (0.0022) | 0.717 | 1.44 | 1.00 | 0.0111 | (0.90) | (1.90) |
| | 2013 | −0.1459 | 0.0422 | 0.4666 | (0.177) | 0.011 | (0.39) | 0.0066 | 0.1408 | (0.0004) | 0.0133 | 0.7017 | (0.0007) | 0.716 | 0.62 | (0.59) | 0.0389 | (0.12) | 0.47 |
| | 2014 | 0.0826 | 0.0415 | 0.4601 | (0.184) | 0.003 | (0.39) | 0.0059 | −0.0714 | 0.0002 | 0.0133 | 0.7017 | 0.0003 | 0.715 | 0.95 | 0.06 | 0.0317 | (0.32) | (0.38) |
| | 2015 | −0.1590 | 0.0405 | 0.4818 | (0.162) | 0.013 | (0.39) | 0.0049 | 0.2717 | (0.0005) | 0.0133 | 0.7017 | (0.0011) | 0.716 | 0.60 | (0.63) | 0.0231 | (0.56) | 0.06 |
| | 2016 | 0.6614 | 0.0504 | 0.5110 | (0.133) | 0.219 | (0.39) | 0.0148 | −0.1599 | 0.0009 | 0.0133 | 0.7017 | 0.0019 | 0.713 | 1.45 | 1.03 | 0.0237 | (0.55) | (1.58) |
| | 2017 | −0.0974 | 0.0516 | 0.4656 | (0.178) | 0.005 | (0.39) | 0.0160 | −0.1599 | 0.0010 | 0.0133 | 0.7017 | 0.0020 | 0.713 | 0.69 | (0.46) | 0.0151 | (0.79) | (0.33) |
| | 2018 | −0.1410 | 0.0666 | 0.4912 | (0.153) | 0.010 | (0.39) | 0.0310 | −0.1599 | 0.0020 | 0.0133 | 0.7017 | 0.0039 | 0.711 | 0.60 | (0.63) | 0.0184 | (0.70) | (0.07) |

**Table A11.** Calculations of Optimal and E×cess Debt: Bank Rakyat Indonesia.

| Bank | Year | Capital Gains/(Losses), (r) | Interest Rate (i) | Beta (Productivity of Capital, β) | Beta Variance (xy(t)) | Half Square of Capital Gain Variance | Correlation of Interest and Capital Gain Variables | Interest Rate Variance | Capital Gain Variance | Correlation and Variances of Interest and Capital Gain | Std. Deviation of Interest Rate | Std. Deviation of Capital Gain | 2 × (Correlation and Variances of Interest and Capital Gain) | Risk | Optimal Debt Ratio, f*(t) | Normalized Optimal Debt Ratio | Actual Debt Ratio | Normalized Actual Debt Ratio | Excess Debt |
|---|---|---|---|---|---|---|---|---|---|---|---|---|---|---|---|---|---|---|---|
| Bank Rakyat Indonesia | 2000 | 0.0000 | 0.0258 | 0.0000 | (0.880) | 0.000 | (0.20) | (0.0098) | 0.3260 | 0.0006 | 0.0133 | 0.4291 | 0.0013 | 0.441 | 1.94 | (0.36) | 0.0605 | 2.17 | 2.53 |
| | 2001 | 0.0000 | 0.0243 | 0.0000 | (0.880) | 0.000 | (0.20) | (0.0113) | 0.0636 | 0.0001 | 0.0133 | 0.4291 | 0.0003 | 0.442 | 1.94 | (0.36) | 0.0484 | 1.44 | 1.80 |
| | 2002 | 0.0000 | 0.0209 | 1.5957 | 0.716 | 0.000 | (0.20) | (0.0147) | 1.2205 | 0.0036 | 0.0133 | 0.4291 | 0.0072 | 0.435 | 1.98 | (0.29) | 0.0220 | (0.15) | 0.14 |
| | 2003 | 0.0000 | 0.0188 | 1.4731 | 0.593 | 0.000 | (0.20) | (0.0168) | −0.1798 | (0.0006) | 0.0133 | 0.4291 | (0.0012) | 0.444 | 1.94 | (0.36) | 0.0244 | (0.01) | 0.35 |
| | 2004 | 1.0869 | 0.0286 | 1.1355 | 0.255 | 0.591 | (0.20) | (0.0070) | −0.0292 | (0.0000) | 0.0133 | 0.4291 | (0.0001) | 0.442 | 3.05 | 1.33 | 0.0279 | 0.20 | (1.13) |
| | 2005 | 0.0082 | 0.0191 | 1.1128 | 0.233 | 0.000 | (0.20) | (0.0165) | 0.2895 | 0.0010 | 0.0133 | 0.4291 | 0.0019 | 0.440 | 1.98 | (0.30) | 0.0254 | 0.05 | 0.36 |
| | 2006 | 0.9014 | 0.0197 | 1.1161 | 0.236 | 0.406 | (0.20) | (0.0159) | −0.2307 | (0.0007) | 0.0133 | 0.4291 | (0.0015) | 0.444 | 3.05 | 1.34 | 0.0189 | (0.34) | (1.68) |
| | 2007 | 0.3794 | 0.0339 | 1.1390 | 0.259 | 0.072 | (0.20) | (0.0017) | −0.6142 | (0.0002) | 0.0133 | 0.4291 | (0.0004) | 0.443 | 2.61 | 0.66 | 0.0174 | (0.43) | (1.09) |
| | 2008 | −0.4869 | 0.0373 | 1.3706 | 0.490 | 0.119 | (0.20) | 0.0017 | 0.8440 | (0.0003) | 0.0133 | 0.4291 | (0.0006) | 0.443 | 0.54 | (2.50) | 0.0071 | (1.05) | 1.45 |
| | 2009 | 0.9983 | 0.0252 | 0.9429 | 0.063 | 0.498 | (0.20) | (0.0104) | 0.0327 | 0.0001 | 0.0133 | 0.4291 | 0.0001 | 0.442 | 3.06 | 1.36 | 0.0126 | (0.72) | (2.08) |
| | 2010 | 0.4497 | 0.0374 | 1.0671 | 0.187 | 0.101 | (0.20) | 0.0018 | −0.5646 | 0.0002 | 0.0133 | 0.4291 | 0.0004 | 0.442 | 2.70 | 0.79 | 0.0060 | (1.12) | (1.91) |
| | 2011 | 0.2627 | 0.0476 | 0.9143 | 0.034 | 0.034 | (0.20) | 0.0120 | 1.1362 | (0.0027) | 0.0133 | 0.4291 | (0.0055) | 0.448 | 2.36 | 0.29 | 0.0046 | (1.20) | (1.49) |
| | 2012 | −0.0229 | 0.0442 | 0.7872 | (0.093) | 0.000 | (0.20) | 0.0086 | 0.3179 | (0.0005) | 0.0133 | 0.4291 | (0.0011) | 0.443 | 1.83 | (0.52) | 0.0040 | (1.24) | (0.71) |
| | 2013 | −0.1741 | 0.0422 | 0.8617 | (0.018) | 0.015 | (0.20) | 0.0066 | 0.1408 | (0.0002) | 0.0133 | 0.4291 | (0.0004) | 0.443 | 1.46 | (1.08) | 0.0100 | (0.88) | 0.20 |
| | 2014 | 0.5733 | 0.0415 | 0.7272 | (0.153) | 0.164 | (0.20) | 0.0059 | −0.0714 | 0.0001 | 0.0133 | 0.4291 | 0.0002 | 0.442 | 2.82 | 0.99 | 0.0147 | (0.60) | (1.58) |
| | 2015 | −0.1277 | 0.0405 | 0.6979 | (0.182) | 0.008 | (0.20) | 0.0049 | 0.2717 | (0.0003) | 0.0133 | 0.4291 | (0.0005) | 0.443 | 1.59 | (0.90) | 0.0393 | 0.89 | 1.79 |
| | 2016 | 0.0452 | 0.0504 | 0.6082 | (0.272) | 0.001 | (0.20) | 0.0148 | −0.1599 | 0.0005 | 0.0133 | 0.4291 | 0.0009 | 0.441 | 1.98 | (0.30) | 0.0411 | 1.00 | 1.30 |
| | 2017 | 0.5534 | 0.0516 | 0.5923 | (0.288) | 0.153 | (0.20) | 0.0160 | −0.1599 | 0.0005 | 0.0133 | 0.4291 | 0.0010 | 0.441 | 2.79 | 0.93 | 0.0380 | 0.81 | (0.12) |
| | 2018 | −0.0530 | 0.0666 | 0.5814 | (0.299) | 0.001 | (0.20) | 0.0310 | −0.1599 | 0.0010 | 0.0133 | 0.4291 | 0.0020 | 0.440 | 1.73 | (0.69) | 0.0441 | 1.18 | 1.87 |

**Table A12.** Calculations of Optimal and E×cess Debt: Panin Bank.

| Bank | Year | Capital Gains/(Losses), (r) | Interest Rate (i) | Beta (Productivity of Capital, β) | Beta Variance (αy(t)) | Half Square of Capital Gain Variance | Correlation of Interest and Capital Gain Variables | Interest Rate Variance | Capital Gain Variance | Correlation and Variances of Interest and Capital Gain | Std. Deviation of Interest Rate | Std. Deviation of Capital Gain | 2 × (Correlation and Variances of Interest and Capital Gain) | Risk | Optimal Debt Ratio, f*(t) | Normalized Optimal Debt Ratio | Actual Debt Ratio | Normalized Actual Debt Ratio | Excess Debt |
|---|---|---|---|---|---|---|---|---|---|---|---|---|---|---|---|---|---|---|---|
| Panin Bank | 2000 | 0.0096 | 0.0258 | 0.4325 | (0.076) | 0.000 | (0.46) | (0.0098) | 0.3260 | 0.0015 | 0.0133 | 0.5636 | 0.0029 | 0.574 | 0.86 | (0.19) | 0.1369 | 3.08 | 3.27 |
| | 2001 | 0.0124 | 0.0243 | 0.6469 | 0.138 | 0.000 | (0.46) | (0.0113) | 0.0636 | 0.0003 | 0.0133 | 0.5636 | 0.0007 | 0.576 | 0.86 | (0.18) | 0.0615 | (0.01) | 0.17 |
| | 2002 | 1.8265 | 0.0209 | 0.8386 | 0.330 | 1.668 | (0.46) | (0.0147) | 1.2205 | 0.0083 | 0.0133 | 0.5636 | 0.0165 | 0.560 | 1.17 | 0.36 | 0.0629 | 0.04 | (0.32) |
| | 2003 | 0.6824 | 0.0188 | 0.5042 | (0.005) | 0.233 | (0.46) | (0.0168) | −0.1798 | (0.0014) | 0.0133 | 0.5636 | (0.0028) | 0.580 | 1.62 | 1.17 | 0.0996 | 1.55 | 0.38 |
| | 2004 | 0.4428 | 0.0286 | 0.4813 | (0.028) | 0.098 | (0.46) | (0.0070) | −0.0292 | (0.0001) | 0.0133 | 0.5636 | (0.0002) | 0.577 | 1.43 | 0.83 | 0.0763 | 0.59 | (0.24) |
| | 2005 | −0.0566 | 0.0191 | 0.5202 | 0.011 | 0.002 | (0.46) | (0.0165) | 0.2895 | 0.0022 | 0.0133 | 0.5636 | 0.0044 | 0.572 | 0.76 | (0.37) | 0.0398 | (0.90) | (0.53) |
| | 2006 | 0.8886 | 0.0197 | 0.4192 | (0.090) | 0.395 | (0.46) | (0.0159) | −0.2307 | (0.0017) | 0.0133 | 0.5636 | (0.0034) | 0.580 | 1.69 | 1.30 | 0.0422 | (0.81) | (2.10) |
| | 2007 | 0.1298 | 0.0339 | 0.3693 | (0.140) | 0.008 | (0.46) | (0.0017) | −0.6142 | (0.0005) | 0.0133 | 0.5636 | (0.0009) | 0.578 | 1.03 | 0.12 | 0.0620 | 0.01 | (0.11) |
| | 2008 | −0.2882 | 0.0373 | 0.6177 | 0.109 | 0.042 | (0.46) | 0.0017 | 0.8440 | (0.0007) | 0.0133 | 0.5636 | (0.0014) | 0.578 | 0.24 | (1.29) | 0.0628 | 0.04 | 1.33 |
| | 2009 | 0.8542 | 0.0252 | 0.4907 | (0.018) | 0.365 | (0.46) | (0.0104) | 0.0327 | 0.0002 | 0.0133 | 0.5636 | 0.0003 | 0.577 | 1.69 | 1.29 | 0.0571 | (0.20) | (1.49) |
| | 2010 | 0.5837 | 0.0374 | 0.4634 | (0.046) | 0.170 | (0.46) | 0.0018 | −0.5646 | 0.0005 | 0.0133 | 0.5636 | 0.0010 | 0.576 | 1.54 | 1.02 | 0.0759 | 0.58 | (0.45) |
| | 2011 | −0.3281 | 0.0476 | 0.4791 | (0.030) | 0.054 | (0.46) | 0.0120 | 1.1362 | (0.0063) | 0.0133 | 0.5636 | (0.0126) | 0.589 | 0.12 | (1.51) | 0.0699 | 0.33 | 1.84 |
| | 2012 | −0.2336 | 0.0442 | 0.4913 | (0.018) | 0.027 | (0.46) | 0.0086 | 0.3179 | (0.0013) | 0.0133 | 0.5636 | (0.0025) | 0.579 | 0.35 | (1.10) | 0.0663 | 0.18 | 1.29 |
| | 2013 | −0.1706 | 0.0422 | 0.5843 | 0.075 | 0.015 | (0.46) | 0.0066 | 0.1408 | (0.0004) | 0.0133 | 0.5636 | (0.0009) | 0.578 | 0.49 | (0.86) | 0.0492 | (0.52) | 0.34 |
| | 2014 | 0.7282 | 0.0415 | 0.5844 | 0.076 | 0.265 | (0.46) | 0.0059 | −0.0714 | 0.0002 | 0.0133 | 0.5636 | 0.0004 | 0.576 | 1.61 | 1.16 | 0.0419 | (0.82) | (1.98) |
| | 2015 | −0.3686 | 0.0405 | 0.4828 | (0.026) | 0.068 | (0.46) | 0.0049 | 0.2717 | (0.0006) | 0.0133 | 0.5636 | (0.0012) | 0.578 | 0.05 | (1.63) | 0.0439 | (0.73) | 0.90 |
| | 2016 | −0.0640 | 0.0504 | 0.4535 | (0.055) | 0.002 | (0.46) | 0.0148 | −0.1599 | 0.0011 | 0.0133 | 0.5636 | 0.0022 | 0.575 | 0.68 | (0.50) | 0.0468 | (0.61) | (0.11) |
| | 2017 | 0.5146 | 0.0516 | 0.4281 | (0.081) | 0.132 | (0.46) | 0.0160 | −0.1599 | 0.0012 | 0.0133 | 0.5636 | 0.0024 | 0.574 | 1.46 | 0.89 | 0.0496 | (0.50) | (1.39) |
| | 2018 | −0.0540 | 0.0666 | 0.3818 | (0.127) | 0.001 | (0.46) | 0.0310 | −0.1599 | 0.0023 | 0.0133 | 0.5636 | 0.0046 | 0.572 | 0.68 | (0.51) | 0.0301 | (1.30) | (0.79) |

**Notes**

[1]  Meaning increases in housing prices.

[2]  If the rate is much lower than the accepted return in the market, say 4–5%, they do not invest/finance. If the rate is high, then they finance but reduce the rent such that the monthly payment would be competitive with that of other banks in the market.

[3]  This source of instability is also discussed by Brunnermeier and Sannikov (2014).

[4]  The full mathematics of the Stein model (Stein 2008, 2012a) are provided in Appendix A.

[5]  The list of banks' names and types is provided in Appendix B.

[6]  The full calculations are presented in Tables A2–A8 in Appendix C.

[7]  Variables in the first two columns, capital gain/loss and interest rate, form the uncertainty of the model. The two variables are stochastic in the model and can move in different directions.

[8]  The reason that total capital is calculated this way is because capital investments in a company are made up of equity capital and debt financing; hence, a company has two types of stakeholders: equity and debt holders.

[9]  In the calculation, this correlation is a constant value over the period.

[10]  The optimal debt ratio is positive only if the net return is greater than the risk premium.

[11]  They were normalized such that each variable had a mean of zero.

[12]  Such as cash, accounts receivable, investments in other firms, properties, and intangible assets.

[13]  See attached Appendix C for detailed calculations of excess debt in Tables A2–A8.

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
