# Peer review of "Financial Crises and Business Cycle Implications for Islamic and Non-Islamic Bank Lending in Indonesia"

_jrfm, doi:10.3390/jrfm15070292_

Round 1
Reviewer 1 Report
It is necessary to make adjustments in the text format (line spacing p. 3.4, etc., paragraphs, spaces in the text and literature). I would recommend extending the conclusions to compare the behavior of the banking sector in other countries.
Author Response
Dear Referee, I appreciate your constructive feedback helping me improve the paper. I have tried my best to incorporate your comments in this new version of the paper (changes are in red color). In what follows I detail some of the changes I have made in response to your comments:
1. It is necessary to make adjustments in the text format (line spacing p. 3.4, etc., paragraphs, spaces in the text and literature).
Response: Formatting and spacing has been adjusted. Also, please note that I have also had our text reviewed and adjusted by professional editorial and proof-reading service.
2. I would recommend extending the conclusions to compare the behavior of the banking sector in other countries.
Response: The below was added. Thank you.
The results of this paper show that Islamic and conventional banks in Indonesia had a consistent behavior with similar banks in Bahrain, Kuwait, Malaysia, the United States, and the United Kingdom (see Issa, 2020).

Reviewer 2 Report
I read the article closely: excellent, well written, most interesting.
The author's conclusions make good common sense to me:
Results showed that all banks became vulnerable to the GFC in 2007-2009, whereas credit build-up, i.e., over-leveraging, started in 2005. However, for most of the banks in the sample, Islamic banks performed better and leveraged less during the crisis, which meant these banks were affected less by the crisis and also made them more resilient to a financial crisis. Nevertheless, Islamic banks still in some cases relied on leveraging and were subject to the second-round effect of the global crisis around the years of 2011-2013, when higher excess-debt values were found for Islamic banks. That could also be due to the fact the financial crisis hit the real economy after 2009 because most Islamic bank investments and contracts were backed by real estate and property as collateral.
In conclusion, we clearly see that conventional banks showed high excess-debt before and during the GFC, whereas Islamic banks maintained a low excess-debt level during and after the crisis. This is due to their lower leverage and higher solvency, which allowed them to meet relatively stronger demand for credit; however, for both types of banks, an increase in excess-debt a couple of years preceding the crisis and/or during the crisis was observed although the level of increase differed among banks. Overall, Islamic banks leveraged less than conventional banks. To maintain high profits, conventional banks relied on loans, especially loans with high risks. This made the banks even more vulnerable due to the real estate boom. In addition, banks increased lending for corporate takeovers and leveraged buyouts, which are highly leveraged transaction loans.
I suggest revised title:
Financial Crises and Business Cycle Implications for Islamic and non-Islamic bank lending in Indonesia
Author Response
Dear Referee, I appreciate your constructive feedback helping me improve the paper. The title was changed as per your suggestion. A revised version is attached, and the changes are in Red.
Also, please note that I have also had our text reviewed and adjusted by professional editorial and proof-reading service.
Thank you for your feedback.
Samar

Reviewer 3 Report
This paper studies the excess-debt and the GFC periods for the Indonesian banks. My major comments are listed below.
1) Although some recent references are cited to be motivations of this paper, all methodological literature is rather dated. I would like to see more about recent developments in the excess-debt methods, and how such novel approaches may impact the results of this paper.
2) The GFC is almost 15 years ago. The sample is up to 2018, which is also not novel. I would like to understand more about why the recent COVID crisis is not analyzed, or at least be compared with the GFC.
3) Either from the methodological side or the empirical side, is there any possibility to check the robustness of the baseline results presented in this paper?
Author Response
Dear Referee, I appreciate your constructive feedback helping me improve the paper. I have tried my best to incorporate your comments in this new version of the paper (changes are in red color).
Also, please note that I have also had our text reviewed and adjusted by professional editorial and proof-reading service.
In what follows I detail some of the changes I have made in response to your comments:
1) Although some recent references are cited to be motivations of this paper, all methodological literature is rather dated. I would like to see more about recent developments in the excess-debt methods, and how such novel approaches may impact the results of this paper.
Response: I incorporated four new references that calculate excess debt in page 7. Please see below for your convenience.
This paper focuses on the Stein (2006, 2012) model. However, several recent literatures develop models to estimate excess-debt, for both private and public debt, and many of them stem from the same (2006, 2012) Stein model.
For instance, Issa and Gevorkyan (2022) develop an empirical model of corporate capital structure, optimal debt, and overleveraging, covering approximately the period from 2000 to 2018, across the following six industries: technology, financial, pharmaceutical, auto, airline, and energy. The authors estimate optimal and excess debt for each of the 89 firms and find that corporate excess debt has largely been moving up, spiking around the GFC and continuing into recovery more recently. The trend is consistent with an increase in the actual debt, with varying average excess debt ratios by sector.
An IMF paper by Eberhardt and Presbitero (2018) develops an empirical model using regression analysis to predict banking crises. The authors employ a sample of 60 low-income countries for the time span 1981–2015 and show that credit growth cannot be considered a main driving force of economic distress or financial crisis, as most literature states. The reason is that credit growth is mediated through capital inflow, which is also fueled by a booming financial market. Therefore, banking crises are potentially driven by commodity price changes according to the authors.
Another paper by Nyambuu and Bernard (2015) uses the Stein (2006) model to calculate optimal debt for developing countries. In contrast to this paper, which calculates the optimal debt at the micro-level for the baking sector, the authors apply the model at a macro-level. In both cases, the optimal debt ratios are defined as the threshold beyond which it becomes risky to borrow, or in other words, the distance-from-default indicator variable. While both papers calculate optimal debt, each methodology applies different metrics, since this paper addresses banking default while Nyambuu and Bernard (2015) report sovereign default. Nyambuu and Bernard (2015) show that rising ratios of external debt can make a country vulnerable to shocks and increases the risk of default. Similarly, this paper shows that an increase in excess debt of a certain bank will increase its risk to default on its debt.
Finally, Gevorkyan and Semmler (2016) examine the U.S. shale energy sector, and their model is consistent with Stein’s optimal debt argument as well. The framework is rooted in the theory that the boom trend in borrowing leads to overleveraging and risks of insolvency. Employing a nonlinear model predictive control (NMPC) analysis, the authors present two mutually exclusive scenarios: oligopoly or extensive competition. The empirical analysis of actual and optimal debt based on a vector error correction model (VECM) suggests that firms with larger market capitalization in the industry are more resilient to oil price volatility than medium and small-sized firms.
Gevorkyan, A., & Semmler, W. (2016). Oil price, overleveraging and shakeout in the shale energy sector—Game changers in the oil industry. Economic Modelling, 54, 244-259.
Nyambuu, U., and L. Bernard. (2015). A Quantitative Approach to Assessing Sovereign Default Risk in Resource-Rich Emerging Economies. International Journal of Finance & Economics 20(3): 220–241.
Eberhardt, M. M., & Presbitero, A. (2018). Commodity Price Movements and Banking Crises. International Monetary Fund.
Issa, S., & Gevorkyan, A. V. (2022). Optimal corporate leverage and speculative cycles: an empirical estimation. Structural Change and Economic Dynamics.
2) The GFC is almost 15 years ago. The sample is up to 2018, which is also not novel. I would like to understand more about why the recent COVID crisis is not analyzed, or at least be compared with the GFC.
Response: COVID-19 crisis is not analyzed to the nature of the crisis due to an external shock, while the GFC was a result of a vulnerable state of the financial system. I added some explanation below which was added to the paper.
It is important to differentiate between the nature of the external COVID-19 shock and the 2008 crisis. The former is a purely exogenous shock while the latter is a result of a vulnerable state of the financial system. COVID-19 provides a shock that is not connected to business or financial cycles and, as such, behavior can be analyzed empirically with less concern for the usual endogeneity issues. It must be noted that the government should control leverage in order to prevent a future crisis of any type (see what happened in COVID-19). The COVID-19 pandemic delivered a structurally different impact, but still layering upon subdued endogenous fragilities. For the main endeavor of this paper, this paper proceeds with the core Stein model as laid out in Equations (1) and (2), leaving the COVID-19 crisis and the extension opportunities for future projects.
The below was added to the conclusion:
As seen above, the excess leverage taken by the banking sector in Indonesia poses serious threats. The unprecedented policy response to the COVID-19 pandemic has helped prevent a meltdown and maintain the flow of credit to the economy. However, the outlook remains highly uncertain and vulnerabilities are rising, representing a potential early warning signal.
3) Either from the methodological side or the empirical side, is there any possibility to check the robustness of the baseline results presented in this paper?
Response:
The paper does not use econometrics and uses an application of a mathematical equation. Therefore, it’s not possible to perform any robustness checks.

Round 2
Reviewer 3 Report
Thank you for addressing my comments. I recommend publication as in its current form.